# Structural basis for human chondroitin sulfate chain polymerization

Poushalee Dutta [1], Rosa L. Cordeiro[1], Mélanie Friedel-Arboleas[1], Marie Bourgeais[1], Sylvain D. Vallet [1], Margot Weber[1], Margaux Molinas[1], Huazhang Shu [2], Magnus N. N. Grønset [2], Rebecca L. Miller [2], Elisabetta Boeri Erba [1] & Rebekka Wild [1] ✉

Chondroitin sulfates are complex polysaccharide chains that regulate various biological processes at the cell surface and within the extracellular matrix. Here, we identify four heterodimeric complexes responsible for chondroitin sulfate chain polymerization in humans: CHSY1-CHPF, CHSY1-CHPF2, CHSY3-CHPF, and CHSY3-CHPF2. Using a custom-tailored in vitro glycosylation assay based on chemo-enzymatically synthesized fluorescent substrates, we demonstrate that all four complexes exhibit chain polymerization activity. The cryo-EM structure of the CHSY3-CHPF complex provides molecular insights into the chondroitin sulfate chain polymerization reaction. The architecture of the catalytic sites suggests that CHSY1 and CHSY3 are enzymatically active, while CHPF and CHPF2 primarily play a stabilizing role. Mutational analysis of purified enzyme complexes, combined with an in cellulo complementation assay, confirms that only CHSY1 and CHSY3 have bifunctional glycosyl-transferase activities. Based on the spatial arrangement of the catalytic sites, we propose that chondroitin sulfate chain polymerization follows a non-processive, distributive mechanism.

Chondroitin sulfates (CS) are linear, negatively charged, and complex polysaccharide chains ubiquitously expressed on the cell surface and within the extracellular matrix of almost all higher eukaryotes. The polysaccharide backbone consists of repeating disaccharide units of N-acetyl-D-galactosamine (GalNAc) and D-glucuronic acid (GlcA), and carries sulfate groups at various positions. CS chains are covalently bound to specific serine residues of core proteins, together forming chondroitin sulfate proteoglycans (CSPGs)[1,2]. CSPGs play a key role in regulating a broad range of cellular processes, including cellular signaling, morphogenesis, extracellular matrix remodeling, and neuronal plasticity[3–6]. This extraordinary functional versatility is associated with the structural diversity of CS chains, triggering interactions with a wide range of bioactive molecules[7,8]. Defects in the CS biosynthesis machinery have been linked to osteoarthritis, inflammation, pathogen infections, and various cancers[9–12].

Biosynthesis of CS occurs within the Golgi-lumen and is initiated by the addition of a tetrasaccharide linker composed of glucuronic acid-galactose-galactose-xylose (GlcAβ1–3Galβ1–3Galβ1–4Xyl)[13,14]. This linker is common to various glycosaminoglycans, and it is the subsequent addition of a GalNAc residue by CSGALNACT1 or CSGAL-NACT2 that directs the synthesis toward a CS chain. Accordingly, the enzymes responsible for adding the fifth sugar residue—CSGALNACT1/CSGALNACT2 or EXTL3—determine whether a CS or heparan sulfate chain is synthesized, respectively[13–17]. Following chain initiation, the chondroitin sulfate backbone, consisting of repeating disaccharide units (−4GlcAβ1–3GalNAcβ1-), is polymerized. This backbone then undergoes further modifications, including 4-O- and 6-O-sulfation of GalNAc and 2−O-sulfation of GlcA, to complete CS biosynthesis[18,19]. Additionally, the CS backbone can be modified by glucuronyl C5-epimerase, which converts GlcA to iduronic acid, leading to the formation of dermatan sulfate[1,2].

[1]Institut de Biologie Structurale, UMR 5075, University Grenoble Alpes, CNRS, CEA, Grenoble, France. [2]Department of Cellular and Molecular Medicine, Faculty of Health Sciences Copenhagen Center for Glycocalyx Research, University of Copenhagen, Copenhagen N, Denmark. ✉e-mail: rebekka.wild@ibs.fr

Four human genes—*CHSY1* (Q86X52), *CHSY3* (Q70JA7), *CHPF* (Q8IZ52), and *CHPF2* (Q9P2E5)—have been associated with CS chain polymerization. Sequence analysis suggests that these four proteins, hereafter referred to as CS synthases, are type II membrane proteins with a shared topology: a short N-terminal cytoplasmic tail, followed by a single transmembrane helix, a flexible stem loop, and a large Golgi-luminal catalytic domain[20]. The catalytic domain is predicted to contain an N-terminal GT domain belonging to the GT31 family, which, according to the Carbohydrate-Active Enzymes (CAZy) database, includes diverse inverting β−1,3-glycosyltransferases (GT), and a C-terminal GT domain classified within the GT7 family, which

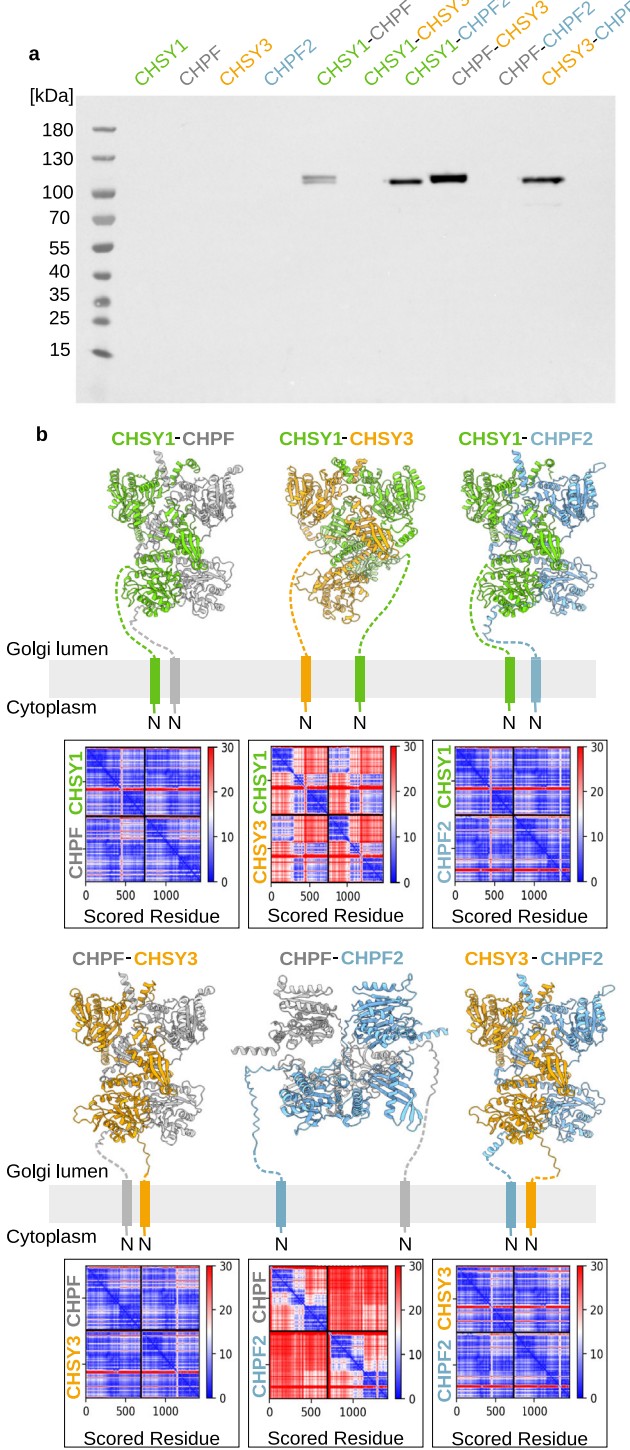

**Fig. 1 | CS synthases can form heterodimeric complexes. a** Transient single- and co-expression of secreted CS synthase-encoding constructs in human embryonic kidney Freestyle 293-F cells. Proteins carrying an N-terminal secretion signal, SUMO (Small Ubiquitin-like Modifier) tag, and 6x His tag were detected in the cell culture medium by western blotting using an anti-His-HRP antibody. Expected molecular weights are 103 kDa for secreted CHSY1 (aa68-aa802), 98 kDa for CHPF (aa81-aa775), 102 kDa for CHSY3 (aa157-aa882), and 101 kDa for CHPF2 (aa57-aa772). Experiment was performed in duplicate (see Supplementary Fig 1b). Source data are provided as a Source Data file. **b** Structural models for the six potential heterodimeric CS polymerase complexes were predicted using AlphaFold 2[33,34]. Models are shown in cartoon representation with CHSY1 in green, CHPF in gray, CHSY3 in orange, and CHPF2 in light blue. The N-terminal anchoring helices and flexible stem regions, which were omitted during model prediction, were drawn by hand. The corresponding predicted aligned error matrices are shown below. AlphaFold 2-predicted models are provided as a Source Data file.

comprises inverting β−1,4-*N*-acetylgalactosaminyltransferases with a GT-A fold[21–23].

CHSY1 and CHSY3 (Chondroitin Sulfate Synthase-1 and −3) have been reported to exhibit both GlcA-transferase (GlcA-T) and GalNAc-transferase (GalNAc-T) activities, with both GT reactions being metal ion-dependent and showing a preference for $Mn^{2+}$ over $Mg^{2+}$ [24,25]. In contrast, Chondroitin Polymerizing Factor (CHPF), formerly known as ChPF[26] or Chondroitin sulfate synthase 2[27], has been suggested to enhance CHSY1 activity rather than catalyzing chain elongation by itself. Lastly, there is CHPF2 Chondroitin Polymerization Factor-2 (CHPF2), which was previously also called CSGlcA-T[21,28]. Analogous to the heterodimeric heparan sulfate polymerase EXT1-EXT2[29,30], previous studies suggested that co-expression of CS synthases enhances CS biosynthesis[28,31].

Based on pull-down experiments, Izumaka and co-workers proposed that heterodimeric complexes can form between all possible combinations of the four CS synthase proteins: CHSY1-CHPF, CHSY3-CHPF, CHSY1-CHSY3, CHSY1-CHPF2, CHSY3-CHPF2, and CHPF-CHPF2[28,31]. In contrast, Petit and colleagues reported that CHSY1 and CHSY3 interact with either CHPF or CHPF2, but not with each other[32]. More recently, Sammon and coworkers described that only the CHSY3-CHPF complex could be produced in a soluble form[13]. These conflicting findings highlight the need to clarify which heterodimeric complexes actually form in cells. Moreover, the functional necessity of having four distinct CS synthases remains unresolved.

In this study, we investigate the formation of heterodimeric CS polymerase complexes by combining co-expression experiments with in silico analysis. Cryo-electron microscopy (cryo-EM) structure determination reveals the architecture of the CHSY3-CHPF CS polymerase complex, which consists of four GT domains. Through extensive in vitro and *in cellulo* mutagenesis studies, we provide detailed molecular insight into CS chain polymerization, unveiling the functional role of the individual CS synthases.

## Results

### CS synthases assemble into four distinct CS polymerase complexes

We set out to investigate whether CS synthases can assemble into homo- and/or heterodimeric complexes, in order to identify the functional CS polymerase unit in human cells. Co-expression studies were performed in human embryonic kidney Freestyle 293-F cells. Single transfections did not result in detectable amounts of proteins, indicating that the proteins were not expressed and secreted in monomeric or homodimeric forms (Fig. 1a). Interestingly, protein expression was observed for four out of the six possible heterodimeric protein combinations: CHSY1-CHPF, CHSY1-CHPF2, CHSY3-CHPF and CHSY3-CHPF2 (Fig. 1a). We further investigated intracellular protein expression to determine whether the absence of secreted protein was due to a lack of expression or inefficient secretion. In addition to the

correctly secreted heterodimeric complexes (CHSY1–CHPF, CHSY1–CHPF2, CHSY3–CHPF, and CHSY3–CHPF2), we observed bands indicative of intracellular CHPF and CHPF2 expression (Supplementary Fig. 1a, b). A possible explanation for the intracellular retention of CHPF/CHPF2 when expressed individually is that they either fail to pass endoplasmic reticulum (ER) quality control or form heterodimeric complexes with endogenous, Golgi-localized CHSY1 or CHSY3. Additional co-expression experiments in HeLa cells confirmed that the formation of the four heterodimeric complexes is not cell-type specific (Supplementary Fig 1c). Similar results were obtained in Chinese hamster ovary (CHO) cells, although protein expression in this system was near the detection limit (Supplementary Fig 1d).

To further investigate the formation of potential homo- and heterodimeric CS polymerase complexes, structural models for all possible combinations were predicted using AlphaFold 2[33,34](Fig. 1b and Supplementary Fig. 2). Consistent with the co-expression experiments, in silico analysis suggested the formation of the heterodimeric complexes CHSY1-CHPF, CHSY1-CHPF2, CHSY3-CHPF and CHSY3-CHPF2 with high confidence, as indicated by low Predicted Aligned Error (PAE) values (Fig. 1b). This high confidence in the heterodimeric complex structure was further supported by interface predicted template modeling (ipTM) values > 0.9 and predicted local distance difference test values around 89 (Supplementary Table 1). Detailed analysis of the structural models revealed tightly packed complexes without major clashes, as well as the presence of a long coiled-coil helix that contributes to the strong interaction between proteins. Predictions for CHSY1-CHSY3 and CHPF-CHPF complexes resulted in models with moderate confidence, whereas models for CHSY1-CHSY1, CHSY3-CHSY3, and CHPF2-CHPF2 had the lowest confidence based on low ipTM values (<0.4) and high distances in PAE plots (Fig. 1b and Supplementary Fig. 2, Supplementary Table 1).

Based on our co-expression and in silico experiments, we propose the existence of four heterodimeric complexes out of the six possible combinations. Interestingly, we always found either CHSY1 or CHSY3 to interact with either CHPF or CHPF2, whereas complexes between CHSY1 and CHSY3 or CHPF and CHPF2 were not observed.

## Purified CS polymerase complexes show CS chain elongation activity

To further investigate the importance of heterodimer formation between CS synthases and to study their differential activities, we purified the four different complexes. We routinely obtained 300 μg to 1 mg of protein complexes from 600 mL Freestyle 293-F suspension culture at satisfying purity, as confirmed by SDS-PAGE analysis (Fig. 2a). To determine the stability of complexes, thermal unfolding was analyzed using nano-differential scanning fluorimetry (nanoDSF). Melting temperatures varied between 43 °C and 48 °C, with the highest stability observed for CHSY3-CHPF and the lowest for the CHPF2-containing complexes CHSY1-CHPF2 and CHSY3-CHPF2 (Fig. 2b and Supplementary Fig. 3a). Next, we performed mass photometry experiments to analyze the oligomeric state of purified complexes in solution. The predominant peaks were found between 166 and 178 kDa, suggesting that the majority of complexes were present in a dimeric form (Fig. 2c–f and Supplementary Table 2). Mass differences of up to 11 kDa between measured and theoretically calculated molecular weights (MW) can be attributed to N-linked glycosylation, as well as to the precision limits of this technique. Further mass photometry experiments demonstrated that the heterodimeric CHSY3–CHPF complex dissociates into monomeric proteins under denaturing conditions (Supplementary Fig 3b, Supplementary Table 3). In order to test if purified CS complexes were functional, we established a Fluorophore-Assisted Carbohydrate Gel Electrophoresis (FACE) assay to efficiently follow CS chain polymerization in vitro. Expanding a chemo-enzymatic synthesis method developed by us and others[13,14], we generated custom-tailored fluorescent glycopeptides, which

served as acceptor substrates for the CS polymerase complex. The selected peptide sequence EEASGEAS was derived from the chondroitin sulfate proteoglycan Colony Stimulating Factor 1 (CSF1)[35], and was chemically synthesized harboring an N-terminal TAMRA fluorophore. Using recombinantly expressed and purified enzymes carrying out glycosaminoglycan biosynthesis, a pentasaccharide linker (GalNAc-GlcA-Gal-Gal-Xyl) was covalently added onto the CSF1 peptide (Supplementary Fig. 4a–c). The resulting Penta-CSF1 peptide was characterized by mass spectrometry (MS) (Supplementary Fig. 5). All four purified CS polymerase complexes were able to elongate the CS chain, using Penta-CSF1 as acceptor substrate and the uridine diphosphate (UDP)-sugars UDP-GlcA and UDP-GalNAc as donor substrates, with the CHSY3-CHPF2 complex generating slightly shorter CS chains (Fig. 3a, b). Treatment of the reaction product with chondroitinase ABC confirmed that the observed band shift was due to chondroitin backbone polymerization (Fig. 3c). To estimate the length of the synthesized chondroitin chains, reaction products were analyzed by size-exclusion chromatography and compared to commercial fluorescein isothiocyanate (FITC)-labeled dextran standards. Chondroitin generated by the CHSY3-CHPF2 showed an approximate molecular weight of 43 kDa. The reaction products of the other CS polymerase complexes were estimated to have MW of approximately 137-151 kDa. However, since they eluted near the void volume (V0), their actual size might be larger (Supplementary Fig. 6). These MW correspond to chains of 800 or more glycan units, suggesting that—except for CHSY3-CHPF2—the CS polymerase complexes can efficiently elongate CS chains in vitro until nearly all substrates are consumed.

## Structural insight into CS chain elongation

After demonstrating that purified CS polymerase complexes were well-folded and functional, we sought to determine the 3D structure of a CS polymerase complex using single-particle cryo-EM. The CHSY3-CHPF complex was selected for structural and functional studies due to its highest thermal stability and in vitro chain polymerization activity in our setup. An octasaccharide, (GalNAc-GlcA)$_4$, representing both an acceptor substrate for the GalNAc-T reaction and the product of the GlcA-T reaction, was added to the sample, along with the donor substrate analog/reaction product UDP, prior to EM grid preparation. Cryo-EM analysis provided a 3D reconstruction at an estimated nominal resolution of 3.0 Å, revealing the overall architecture of the CS polymerase complex with its four GT domains (Fig. 4a, b, Supplementary Fig. 7 and 8, Supplementary Table 4). The experimental structure closely aligns with the AlphaFold 2-predicted model, showing a root-mean square deviation of 1.080 Å across 522 pruned atom pairs (Cα atoms). Predicted C-terminal α-helices of both proteins were not visible in the cryo-EM map, probably due to a high conformational flexibility (Supplementary Fig. 9). The analysis revealed that CHSY3 and CHPF form a tightly packed complex, with their respective N-terminal and C-terminal GT domains interacting strongly, resulting in a total interaction surface of 6711 Å$^2$, as determined using PDBePISA[36].

Both proteins harbor a middle domain, which is stabilized through β-strand swapping between CHSY3 and CHPF, and that forms extensive bonds between the two proteins (Supplementary Fig. 10a). These middle domains are unique among GT structures, with the exception of the GalNAc transferases CSGALNACT1 and CSGALNACT2 (Supplementary Fig. 10b), which are also members of the GT7 CAZy family and participate in chondroitin sulfate biosynthesis[23]. Structural similarity searches for the middle domains using the DALI server[37] (Supplementary Tables 5 and 6) and Foldseek-Multimer[38] (Supplementary Table 7) primarily identified cysteine protease inhibitors and affirmers as the closest structural homologs. However, the low sequence identity and lack of conserved protein-binding residues argue against a shared biological function. Based on the current data, the middle domains likely serve primarily to stabilize the CS polymerase complex.

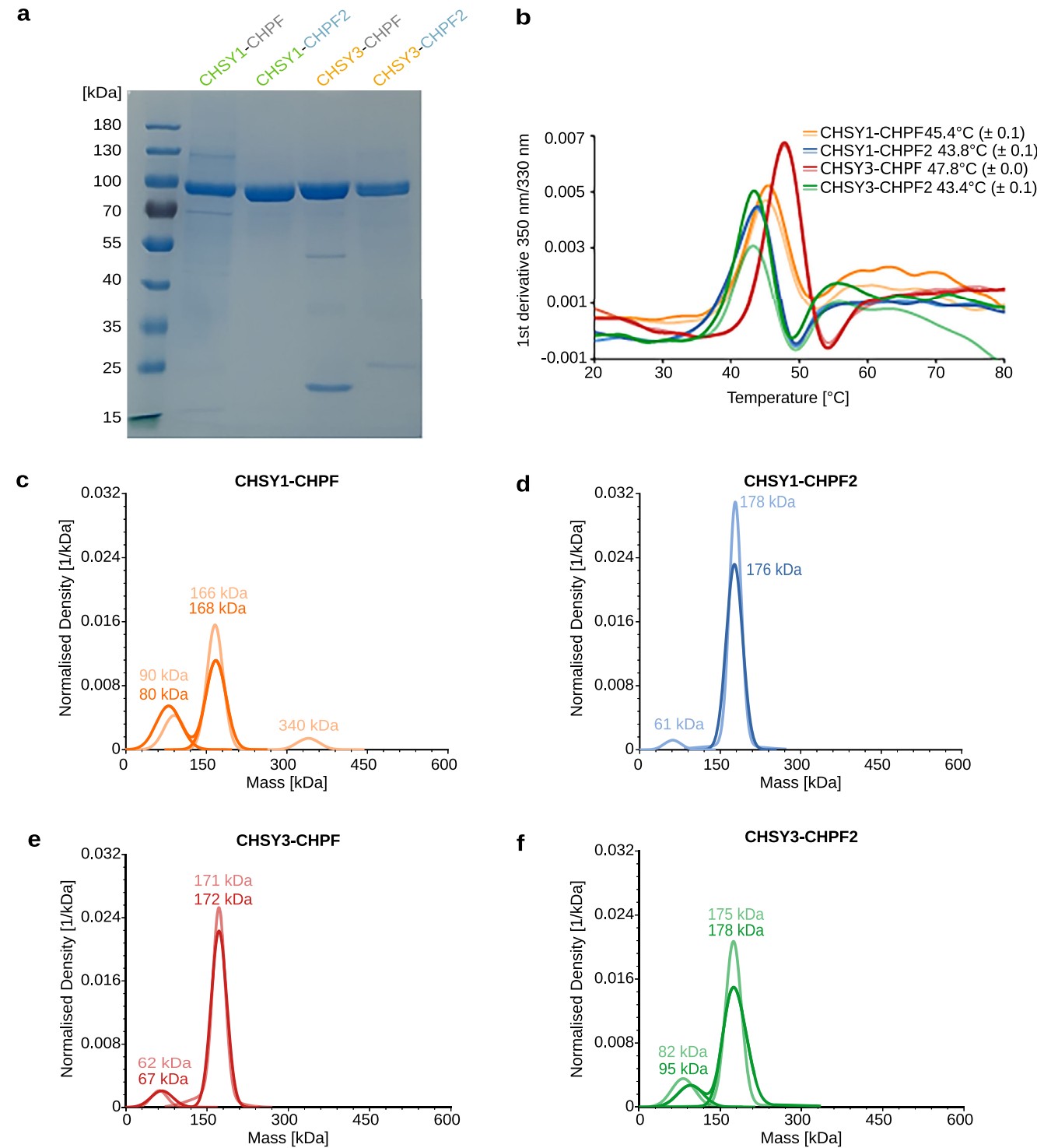

**Fig. 2 | Purified CS polymerase complexes are well-folded. a** Coomassie-stained SDS-PAGE analysis of purified CS polymerase complexes. The expected molecular weight of 3C protease-cleaved proteins is 88 kDa for CHSY1 (aa68-aa802), 86 kDa for CHSY3 (aa157-aa882), 79 kDa for CHPF (aa81-aa775), and 81 kDa for CHPF2 (aa57-aa772). Comparable results were obtained from at least two independent purifications for each complex. **b** The melting temperature ($T_m$) of CS polymerase complexes was determined using nano-differential scanning fluorimetry (nanoDSF). The graph corresponds to the first derivative plot of the fluorescence 350 nm/330 nm ratio. Average melting temperatures from technical duplicate measurements are indicated, with corresponding standard deviation in brackets. **c–f** Mass photometry analysis of purified CS polymerase complexes. The expected molecular masses are 167 kDa for CHSY1–CHPF, 169 kDa for CHSY1–CHPF2, 165 kDa for CHSY3–CHPF, and 167 kDa for CHSY3–CHPF2. Molecular masses from duplicate experiments are shown in the graph; additional measured values are provided in Supplementary Table 2. Source data are provided as a Source Data file.

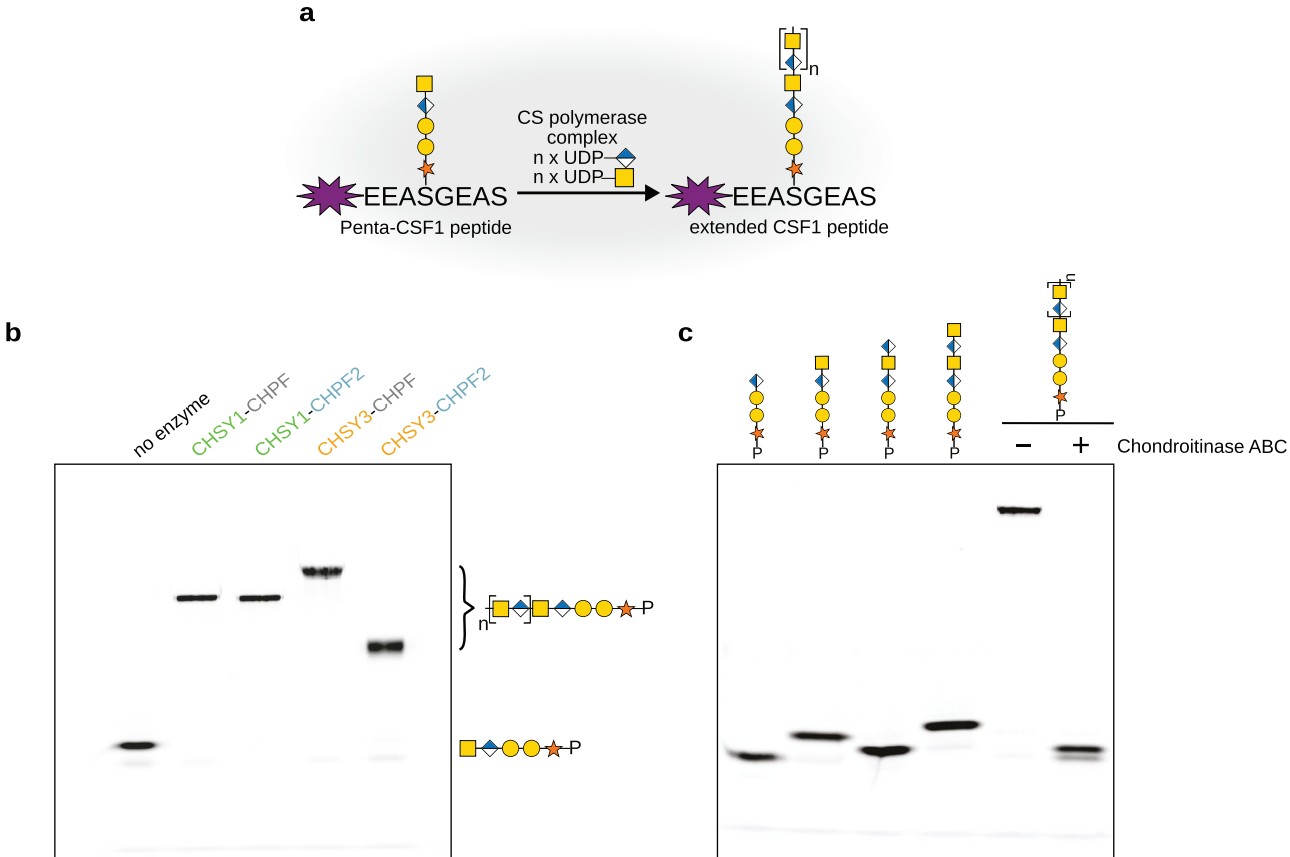

**Fig. 3 | In vitro chain elongation assay using purified CS polymerase complexes. a** Schematic illustration of an in vitro glycosylation assay used to monitor chain elongation by the CS polymerase complexes. The fluorescently labelled Penta-CSF1 peptide, derived from the CSPG colony-stimulating factor 1, served as the acceptor substrate, while UDP-GlcA and UDP-GalNAc were used as donor substrates. Monosaccharide symbols follow the SNFG (Symbol Nomenclature for Glycans) system[67]. **b** Catalytic activity of purified CS polymerase complexes was studied using the in vitro glycosylation assay described in (**a**). Reaction products were separated by polyacrylamide gel electrophoresis, and fluorescent peptides were visualized using a fluorescence imager. This result is representative of three independent experiments. **c** Analysis of CHSY3-CHPF reaction product upon chondroitinase ABC treatment from a single experiment. Tetra-, penta-, hexa-, and heptasaccharide peptides were loaded alongside as length markers. Source data are provided as a Source Data file.

Additionally, EM densities corresponding to three *N*-glycans were identified at asparagine residues N279 and N710 in CHSY3, and N138 in CHPF, with a fourth potential site at N361 in CHPF (Supplementary Fig. 11). Furthermore, EM density was observed for a UDP molecule and an $Mn^{2+}$ ion in the active site of the N-terminal GT domain of CHSY3, as well as for an $Mn^{2+}$ ion in the C-terminal GT domain of CHSY3. No EM density was observed for the octasaccharide substrate that was added during sample preparation.

To gain insights into the catalytic mechanism of CS chain polymerization, we analyzed the different GT domains in more detail. The N-terminal GT domain of CHSY3 (residues L173-P395) belongs to the GT31 family, which mostly comprises inverting β−1,3-glycosyltransferases[23]. The active site architecture highlights key residues involved in catalyzing CS chain elongation (Fig. 5a). These include a DxD motif (D261 and D263), commonly found in GT-A fold enzymes[39,40], which, together with H394, coordinates the $Mn^{2+}$ ion. Additional residues contributing to donor substrate binding in CHSY3 are D231 and K238, located near the uracil moiety; D262, which interacts with the ribose group; and R187, Y233, and K397, which stabilize the pyrophosphate group. Importantly, these residues are also conserved in CHSY1, while most of them are absent in CHPF and CHPF2 (Fig. 5a). This observation suggests that CHPF and CHPF2 are unlikely to bind the UDP-sugar donor substrate and, consequently, are not expected to catalyze glycan transfer.

The C-terminal domain of CHSY3 (residues K620-S882) belongs to the GT7 family, which groups inverting β−1,4-*N*-acetylgalactosaminyltransferases with a GT-A fold[23]. To identify residues involved in donor substrate binding, we performed a search for structurally related proteins using the protein structure comparison server DALI[37]. The chondroitin polymerase from *Escherichia coli* strain K4 (PDB-ID: 2Z87)[41] was identified as a close structural homolog (Supplementary Fig. 10c). By superimposing its crystal structure, which had been determined in complex with a UDP-GalNAc donor substrate and a $Mn^{2+}$ ion in its active site, we identified key residues in the C-terminal GT domain of CHSY3, including D718 and D720, which form the DxD motif, as well as P626, R630, R697, E806, D807, H831, and H834 (Fig. 5b). As with the N-terminal GT31 domain, the residues identified to contribute to donor substrate binding in the GT7 domain were conserved in CHSY1 and CHSY3, but not in CHPF and CHPF2 (Fig. 5b).

### In vitro characterization of CS chain polymerization

To investigate the catalytic roles of the two putative active sites in CHSY3, we constructed structure-based point mutants targeting these regions. The constructs CHSY3[D261N/D263N] and CHSY3[H394A] carry mutations in the N-terminal GT domain, while CHSY3[D718N/D720N] and CHSY3[H831A] have amino acid substitutions in the C-terminal GT domain. Since CHPF does not harbour a DxD motif, it was kept in its wild-type form for analyzing CS polymerization activity of the CHSY3-CHPF

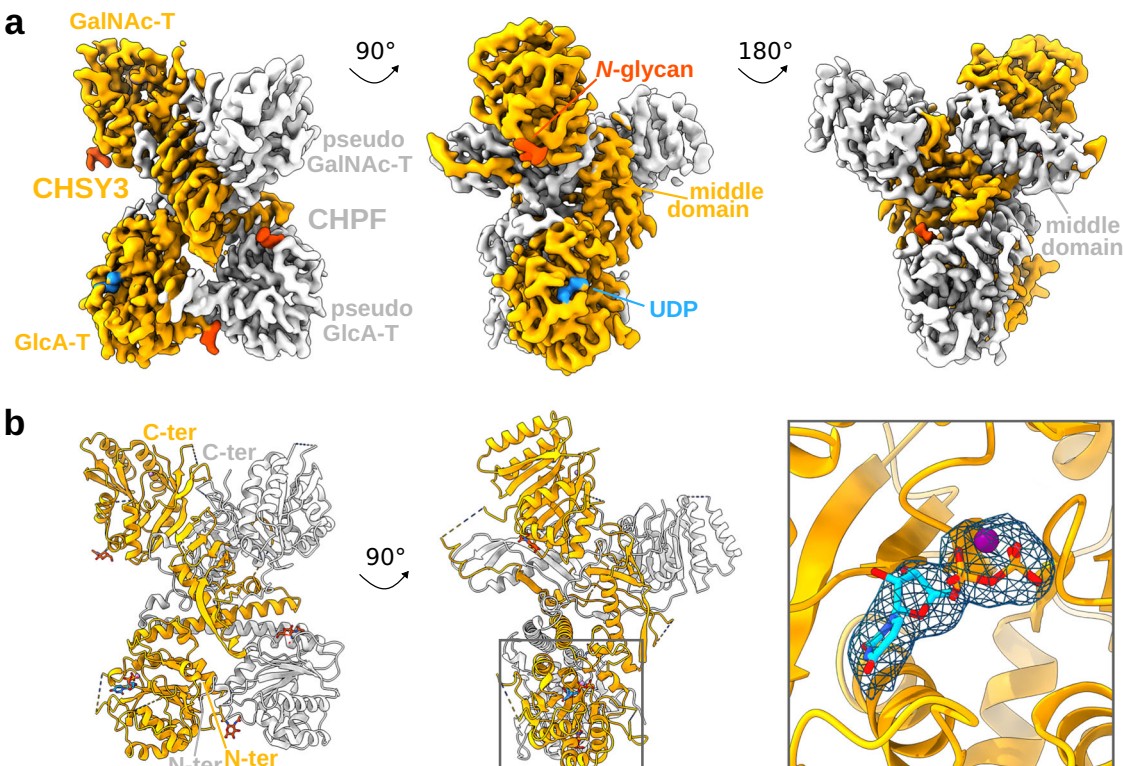

**Fig. 4 | Structure of the human CS polymerase complex CHSY3-CHPF. a** Cryo-EM map of the heterodimeric CHSY3-CHPF complex. Density covering CHSY3 is shown in orange and CHPF in gray. A UDP molecule is highlighted in blue, and *N*-glycans are depicted in dark orange. Both proteins contain an N-terminal GT domain with putative D-glucuronic acid transferase (GlcA-T) activity and a C-terminal GT domain with putative *N*-acetyl-D-galactosamine transferase (GalNAc-T) activity. The complex is shown from three rotated views. **b** Structural model of the complex, shown in cartoon representation with the same color scheme as in (**a**). UDP and *N*-glycans are shown in stick representation. A close-up view shows EM density around UDP and $Mn^{2+}$ ion at a contour level of 0.125.

complex. All four CHSY3$^{mutant}$-CHPF complexes were successfully purified (Supplementary Fig. 12a), and their stability and integrity were confirmed through nanoDSF (Supplementary Fig. 12b, c) and mass photometry analysis (Supplementary Fig. 13 and Supplementary Table 8). The ability of these CHSY3$^{mutant}$-CHPF complexes to perform chain elongation was assessed using the in vitro glycosylation assay.

For this assay, the fluorescent Penta-CSF1 peptide, which carries a pentasaccharide linker (GalNAc-GlcA-Gal-Gal-Xyl), was used to monitor extension with a GlcA unit. Similarly, the Hexa-CSF1 peptide, carrying a hexasaccharide linker (GlcA-GalNAc-GlcA-Gal-Gal-Xyl), was used to follow extension with a GalNAc moiety (Supplementary Figs. 4 and 14).

Upon incubation of the acceptor substrates with the donor substrates UDP-GalNAc and UDP-GlcA in the presence of either wild-type or mutant CHSY3-containing CHSY3-CHPF complexes, the reaction products were separated by gel electrophoresis and visualized by fluorescence imaging. The wild-type CHSY3-CHPF complex efficiently elongated the Penta-CSF1 and Hexa-CSF1 substrates into high molecular weight chains (Fig. 5c, d). In contrast, the CHSY3$^{D261N/D263N}$-CHPF and CHSY3$^{H394A}$-CHPF complexes, which carry mutations in the N-terminal GT domain, were unable to catalyze GlcA transfer (Fig. 5c). Interestingly, incubation of the C-terminal mutants, CHSY3$^{D718N/D720N}$-CHPF and CHSY3$^{H831A}$-CHPF, with both UDP-GlcA and UDP-GalNAc donor substrates resulted into a distinct band shift, indicating the addition of a single GlcA unit to the Penta-CSF1 substrate. This increased migration speed of Hexa-CSF1 compared to Penta-CSF1 reflects the higher negative charge of the elongated product (Fig. 5c). The C-terminal mutants failed to catalyze further chain polymerization, suggesting an inability to transfer GalNAc residues (Fig. 5c). These findings were further confirmed using Hexa-CSF1 as the acceptor

substrate. The N-terminal GT mutants CHSY3$^{D261N/D263N}$-CHPF and CHSY3$^{H394A}$-CHPF catalyzed the transfer of a GalNAc residue, producing the Hepta-CSF1 peptide (Fig. 5d). However, no additional chain elongation was observed for CHSY3$^{D261N/D263N}$-CHPF, validating its impairment in GlcA transfer. In contrast, faint additional bands detected for the CHSY3$^{H394A}$-CHPF complex suggest residual, albeit low, GlcA-T activity. The C-terminal mutants CHSY3$^{D718N/D720N}$-CHPF and CHSY3$^{H831A}$-CHPF failed to elongate the Hexa-CSF1 peptide (Fig. 5d). In summary, the in vitro chain elongation assays revealed that the N-terminal GT domain exhibits GlcA-T activity, while the C-terminal GT domain exhibits GalNAc-T activity. Importantly, although CHPF was maintained in its wild-type form, the mutant CHSY3-CHPF complexes were unable to catalyze chain elongation, suggesting that CHPF itself cannot perform either GlcA or GalNAc transfer.

Analysis of the CHSY3-CHPF complex in the presence and absence of various metal ions revealed that both GlcA-T and GalNAc-T activities are metal ion dependent. The highest catalytic activity was observed with $Mn^{2+}$, consistent with previous reports[27], while intermediate activity was observed with $Mg^{2+}$ and $Ca^{2+}$ (Fig. 5e, f). GlcA-T and GalNAc-T activities were still observed in control reactions without added divalent cations. This finding suggests that the purified enzyme complex already has metal ions bound, likely originating from the buffer used during the initial purification steps.

To confirm that point mutations in the GlcA-T or GalNAc-T catalytic sites did not affect the overall functionality of the CHSY3-CHPF complex, a rescue experiment was designed. The CHSY3$^{D261N/D263N}$-CHPF and CHSY3$^{D718N/D720N}$-CHPF mutants were mixed in equimolar amounts and incubated with Penta-CSF1 peptide, UDP-GalNAc, and UDP-GlcA donor substrates. The reactions were carried out, and the resulting chain polymerization was comparable to that observed for

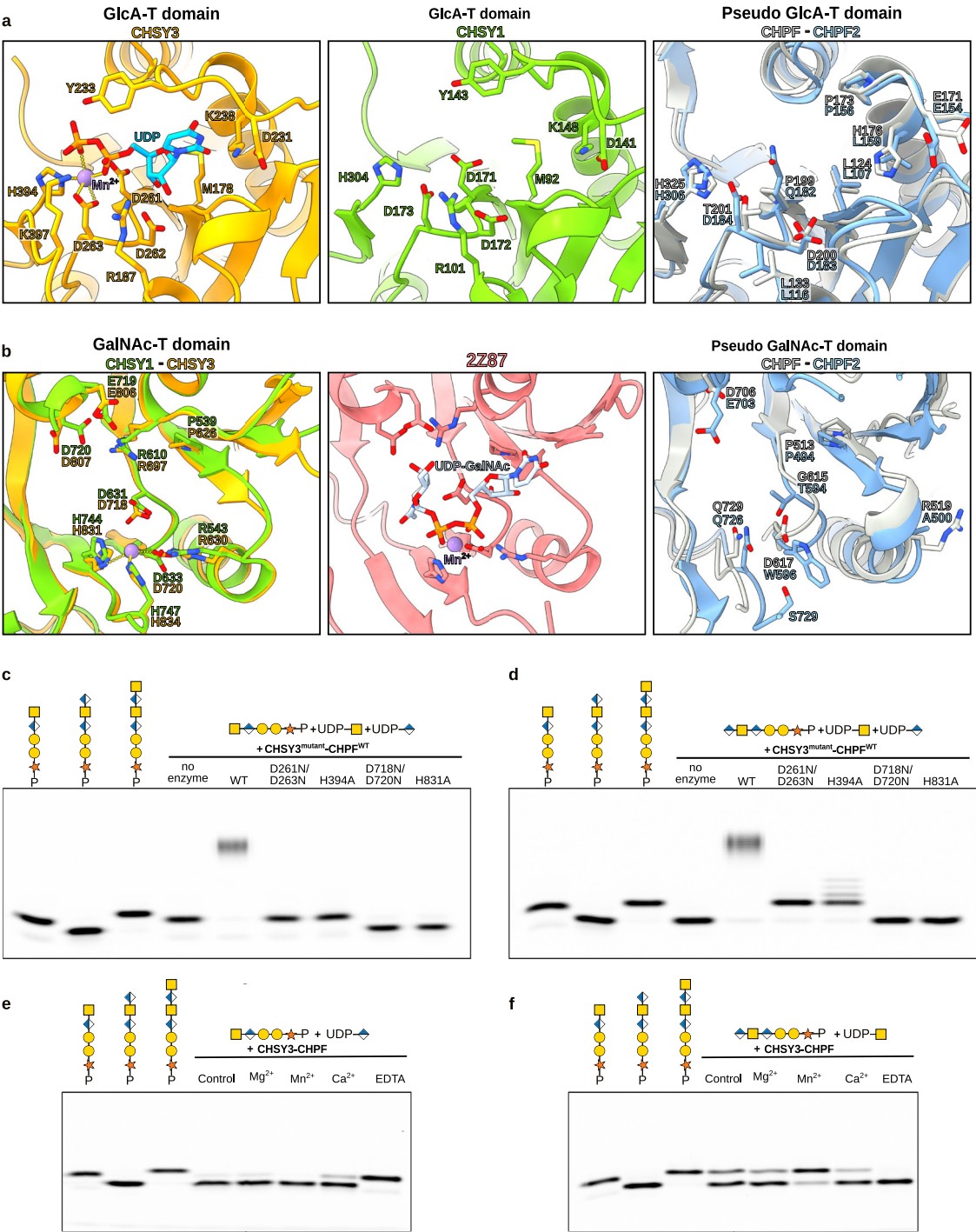

**Fig. 5 | Identification of catalytic residues in the N- and C-terminal GT domains of CS synthases. a, b** Structural comparisons of the putative catalytic sites of the N-terminal GT31 domains and (**b**) the C-terminal GT7 domains of CHSY3 (colored in orange), CHSY1 (colored in green), CHPF (colored in gray), and CHPF2 (colored in blue). CHSY3 and CHPF structures were determined using cryo-EM, while CHSY1 and CHPF2 were predicted using AlphaFold 2[33,34]. **a** A UDP molecule, representing a substrate analog or reaction product, bound to CHSY3, is shown in stick representation and colored in cyan, and the Mn²⁺ ion is shown as a purple sphere. Residues in CHSY3 important for UDP binding are highlighted in stick representation. **b** In the C-terminal GT domain, a structural superimposition with the chondroitin polymerase from *Escherichia coli* strain K4 in complex with UDP-GalNAc (colored in light blue) (PDB-ID: 2Z87)[41] allowed us to propose the interactions with the donor substrate. All the residues identified to interact with the substrate donors are represented in sticks. **c–f** In vitro glycosylation assays for the CHSY3-CHPF complexes, followed by FACE analysis. Images shown are representative of three independent experiments. Source data is provided. **c** Activity of wild-type and CHSY3-mutant complexes, in which mutations are localized in the N-terminal GT domain. The pentasaccharide peptide Penta-CSF1 was used as the acceptor substrate, while both UDP-GalNAc and UDP-GlcA were used as donor substrates. **d** Activity of wild-type and CHSY3-mutant complexes with mutations in the C-terminal GT domain. The hexasaccharide peptide Hexa-CSF1 was used as the acceptor and UDP-GalNAc and UDP-GlcA were used as substrate donors. **e** Effect of different metal ion species on the GlcA and **f** GalNAc transferase reaction. For all the FACE gels, penta-, hexa-, and heptasaccharide peptides were loaded alongside as weight markers. Monosaccharide symbols follow the SNFG (Symbol Nomenclature for Glycans) system[67]. Source data are provided as a Source Data file.

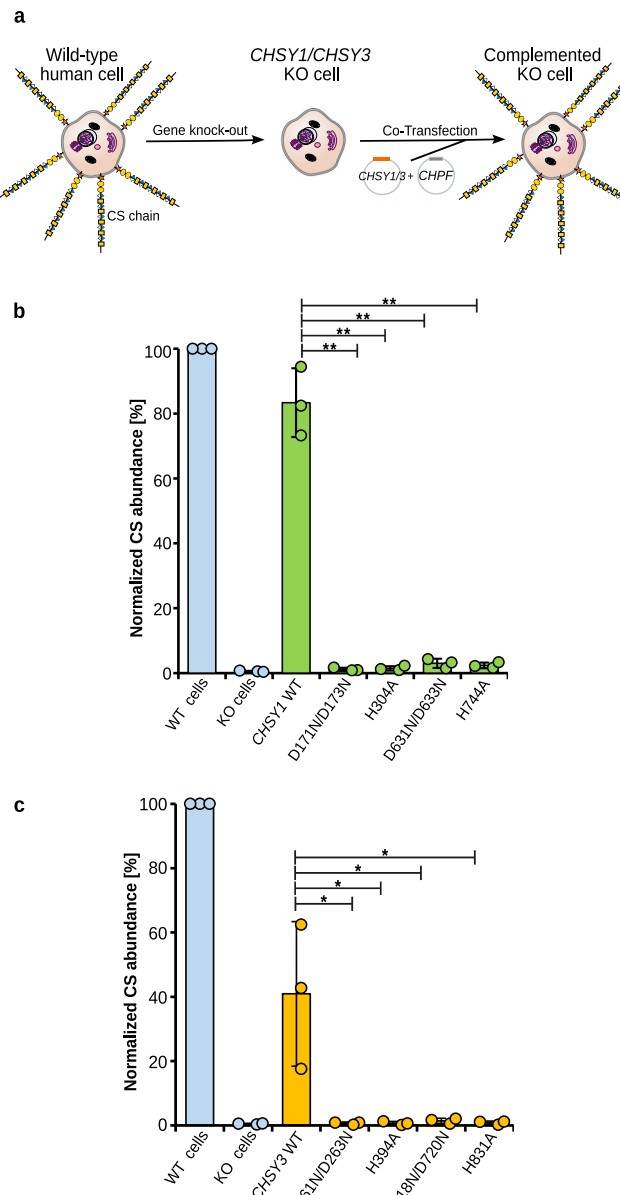

**Fig. 6 | Studying the effects of mutations in the CS polymerase complex on cellular CS biosynthesis. a** Schematic illustration of the *in-cellulo* complementation assay. Knockout (KO) of the *CHSY1* and *CHSY3* genes via CRISPR/Cas eliminates the production of CS chains. KO cells are then co-transfected with either CHSY1 and CHPF or CHSY3 and CHPF encoding plasmids, and cell surface CS levels quantified. **b** Complementation of *CHSY1/CHSY3* KO cells with CHPF wild-type and CHSY1 wild-type or mutant constructs. Cell surface CS levels in non-transfected cells (blue bars) and co-transfected cells (green bars) were quantified by flow cytometry using a primary anti-CS antibody and an Alexa Fluor 488-labeled secondary antibody. Results were compared to wild-type HEK293-6E cells. Experiments were performed in triplicate ($n = 3$). Error bars indicate standard deviation from the mean values. A one-sided paired Student's $t$-test was used to determine $p$-values (*<0.05 and **<0.01) or more precisely CHSY1 WT vs D171N/D173N (0.0030), H304A (0.0030), D631N/D633N (0.0026) and H744A (0.0031). **c** Complementation assay as described in (b), but with CHSY3-CHPF (orange bars). Experiments were performed in triplicate ($n = 3$). Error bars indicate standard deviation from the mean values. A one-sided paired Student's $t$ test was used to determine $p$-values (*<0.05 and **<0.01) or more precisely CHSY3 WT vs D261N/D263N (0.0439), H394A (0.0442), D718N/D720N (0.0438) and H831A (0.0437). Source data is provided. Source data are provided as a Source Data file.

wild-type CHSY3-CHPF complexes (Supplementary Fig. 15). This suggests that the mutations in CHSY3 did not alter the overall stability of the complex. Furthermore, the fact that mixing the GlcA-T and GalNAc-T deficient mutant complexes restored chain polymerization to wild-type levels supports the hypothesis that CS chain elongation follows a distributive rather than a polymerization mechanism, where the polysaccharide chain detaches from the complex before entering the next catalytic cycle.

### Mutations in CHSY1 and CHSY3 impair cellular CS production

Next, we established an *in cellulo* complementation assay to study the effect of structure-based point mutations in CHSY1 and CHSY3 on cellular CS biosynthesis (Fig. 6a). A human embryonic kidney (HEK293-6E) cell line lacking both *CHSY1* and *CHSY3* was chosen as a test system[23]. We co-transfected these *CHSY1/CHSY3* knockout (KO) cells with plasmids encoding either full-length CHSY1 and CHPF or full-length CHSY3 and CHPF. The cell surface CS content was quantified by flow cytometry using a primary anti-CS antibody and an Alexa Fluor 488-labeled secondary antibody. To detect transfection efficiency, cells were permeabilized and labeled with a primary anti-FLAG antibody (all constructions contained an N-terminal FLAG-tag) and a cyanine3 secondary antibody. Approximately 40–50% of the cells were successfully transfected (Supplementary Fig. 16).

While CS biosynthesis was almost abolished in the *CHSY1/CHSY3* KO cell line, CS levels were restored to around 80% and 40% of wild-type cells 36 h post-transfection with CHSY1 wild-type and CHSY3 wild-type encoding plasmids, respectively (Fig. 6b, c). In parallel, we transfected plasmids carrying mutations in the GlcA-T domain (CHSY1[D171N/D173N], CHSY1[H304A], CHSY3[D261N/D263N], CHSY3[H394A]) or the GalNAc-T domain (CHSY1[D631N/D633N], CHSY1[H744A], CHSY3[D718N/D720N], CHSY3[H831A]), while keeping CHPF in its wild-type form. All mutant complexes failed to restore CS levels (Fig. 6b, c). These results suggest that both GlcA-T and GalNAc-T activities are essential for promoting cellular CS biosynthesis. They also demonstrate that both CHSY1-CHPF and CHSY3-CHPF CS polymerase complexes can perform CS chain elongation *in cellulo*.

Our co-expression studies demonstrated that the presence of CHPF or CHPF2 is required for the production of secreted CS polymerase complexes (Fig. 1a). However, CHPF and CHPF2 alone are not sufficient to mediate CS chain elongation, as evidenced by the absence of CS chains in complexes containing mutant CHSY1 or CHSY3 (Fig. 6b, c). To further assess the role of CHPF and CHPF2 in CS biosynthesis, *CHPF/CHPF2* double KO cell lines were generated using CRISPR-Cas technology. Analysis of cell surface CS content revealed an absence of detectable CS (Supplementary Fig. 17 and Supplementary Table 9), supporting the hypothesis that the presence of either CHPF or CHPF2 is essential for the formation of stable and functional CS polymerase complexes. These findings also argue against the formation of a functional CHSY1−CHSY3 complex in the absence of CHPF or CHPF2.

## Discussion

### Heterodimeric CS polymerase complexes

Chain polymerization is a key step in CS biosynthesis, and four genes—*CHSY1, CHSY3, CHPF*, and *CHPF2*—have been shown to be involved in CS chain elongation. However, their individual contributions and relative importance remained unclear. In this study, we demonstrate that human CS synthases form heterodimeric complexes. Through co-expression experiments in human embryonic kidney HEK293-F, HeLa and CHO cells, we observed the formation of four out of the six possible heterodimeric CS polymerase complex combinations: CHSY1-CHPF, CHSY1-CHPF2, CHSY3-CHPF and CHSY3-CHPF2 (Fig. 1a and Supplementary Fig. 1). Interestingly, we always found one of the chondroitin sulfate synthases (CHSY1 or CHSY3) to interact with one of the chondroitin sulfate polymerization factors (CHPF or CHPF2), while neither the chondroitin sulfate synthases nor the chondroitin sulfate

polymerization factors appeared to form complexes among themselves.

This aligns with our observations from double KO cell experiments, where CS production was ablated in human cells lacking either *CHSY1* and *CHSY3* or *CHPF* and *CHPF2* (Fig. 6b and Supplementary Fig. 17).

Of note, CHSY1 and CHSY3 share a high sequence identity (68.14%), as do CHPF and CHPF2 (59.15%), whereas CHSY1 exhibits only 23.73% sequence identity with CHPF and 24.61% with CHPF2 (Supplementary Fig. 18a). We did not detect the expression of single CS synthases or the formation of homodimeric complexes (Fig. 1a, Supplementary Fig. 1). In silico analysis of potential CS polymerase complexes using AlphaFold 2 further supported the formation of the same four complexes (Fig. 1b, Supplementary Fig. 2). Additionally, cryo-EM studies of the CHSY3-CHPF complex revealed that the two proteins form a tightly-packed complex with large interaction surfaces. Importantly, no monomers were observed during cryo-EM data processing. Mass photometry analysis of the purified CS polymerase complexes further confirmed that these are rather stable and do not dissociate (Fig. 2c–f). Together, our results suggest that heterodimeric CS polymerase complexes represent the predominant and functional form responsible for CS chain polymerization in human cells. A similar observation has been made for the human heparan sulfate polymerase EXT1-EXT2[29,30]. Golgi-localized glycosyltransferases frequently form homo- or heterooligomeric complexes, which enhances their catalytic activity, stability and regulates their cellular localization[37,38].

Previous studies hypothesized that CS synthases form heterodimeric complexes[28,31]. However, based on pull-down experiments with secreted proteins in the cell medium, Izumikawa and co-workers proposed that any of these proteins could interact to form such complexes[28]. In contrast, our co-expression studies and in silico analysis indicate that CHSY1-CHPF, CHSY1-CHPF2, CHSY3-CHPF, and CHSY3-CHPF2 are the four primary CS polymerase complexes, showing the highest co-expression and strongest likelihood in AlphaFold2 predictions. We do not observe any evidence for the existence of the heterodimeric complexes CHSY1-CHSY3 or CHPF-CHPF2. This conclusion is further supported by in vitro and *in cellulo* assays, which indicate that only CHSY1 and CHSY3 possess catalytic activity, whereas CHPF and CHPF2 have a structural role that is required for enzymatic activity of CHSY1 and CHSY3 (Figs. 5 and 6).

## Mechanistic insights into CS chain polymerization

In order to learn more about CS chain polymerization at the molecular level, we determined the structure of the CHSY3-CHPF complex using cryo-EM. Both CHSY3 and CHPF consist of two GT domains, bridged by a middle domain, and together form a stable heterodimeric complex. The AlphaFold 2-predicted model fitted well into the experimental EM map. Given the high structural similarity between the four CS polymerase complex models, the AlphaFold 2 predictions are likely to be correct and can thus serve as a valuable tool for future structure-based studies.

UDP, the reaction by-product of the GlcA and GalNAc transferase reaction, was added during cryo-EM sample preparation. We identified one UDP molecule bound in the CHSY3 GlcA-T active site. This finding revealed key residues involved in the GlcA-T reaction. In addition, we inferred important amino acid residues for the GalNAc-T reaction through structural comparisons with the *E. coli* K4 chondroitin polymerase (Fig. 5a, b). The GlcA-T domain of CHSY1 and CHSY3 belongs to the CAZy GT31 family, while their GalNAc-T domain is part of the GT7 family. Both GT7 and GT31 families generate a $\beta$-glycosydic linkage using an inverting mechanism. Based on structural comparison with previously characterized homologs, the residues D252 and D720 in CHSY1, and D342 and D807 in CHSY3, were identified as catalytic base residues that start the reaction cycle[42,43]. A significant C-terminal portion, 48 residues in CHSY3 and 45 residues in CHPF, is not visible in the

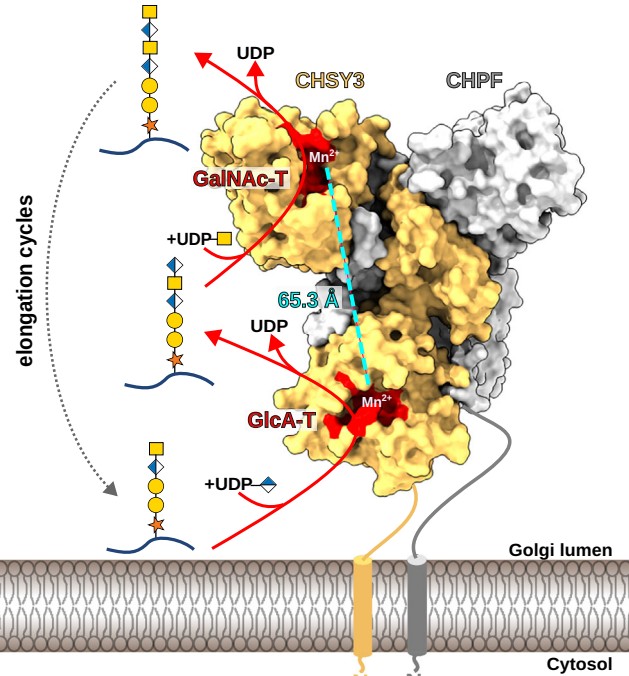

**Fig. 7 | Molecular mechanism of CS chain polymerization.** Illustration of the proposed mechanism for chain elongation by the CS polymerase complex CHSY3-CHPF. In a first reaction step, a GlcA residue is added onto the proteoglycan carrying a pentasaccharide-linker by the N-terminal GT domain of CHSY3. The generated hexasaccharide product is released and binds to the C-terminal GT domain of CHSY3, exhibiting GalNAc transferase activity. Upon GalNAc addition, the proteoglycan dissociates, and a new reaction cycle starts. Repetitive addition of GlcA and GalNAc residues leads to the generation of the long chondroitin sulfate backbone.

cryo-EM structure, likely due to a high flexibility (Supplementary Fig. 9). The AlphaFold 2 model suggests that these regions form α-helices, which are located in close proximity to the active site. The C-termini might thus contribute to acceptor substrate binding, as observed for the GT7 homolog B4GalT7[43]. On the other hand, these helices, when stabilized near the active site, could also block substrate access, thus preventing the GT reaction. Future structural studies on the CS polymerase complex, bound to acceptor substrates of varying length, are needed to characterize acceptor substrate binding in molecular detail. These studies could also reveal an involvement of the flexible C-terminal helices in the transfer reaction.

Based on the distance between the active sites, we propose a distributive rather than a polymerization-based reaction mechanism (Fig. 7). The same observation was made for the bacterial chondroitin polymerase, whose two active sites are located on opposite sides, around 60 Å apart[41]. However, in contrast to human CS biosynthesis, this bacterial chondroitin polymerase functions as a monomer. This raises the question: Why have vertebrates evolved a much more sophisticated CS polymerization machinery that relies on four distinct CS synthases? One possible explanation could be that this redundancy provides a protective mechanism against deleterious mutations. Furthermore, it is conceivable that the ancestral proteins of CHPF (which gave rise to CHPF and CHPF2) and CHSY (which gave rise to CHSY1 and CHSY3)[32] initially possessed catalytic activity. In such a scenario, the formation of heterodimeric complexes would increase the local concentration of enzymes, facilitating chain elongation by reducing the substrate's travel path, enhancing the efficiency of CS biosynthesis.

However, our in vitro and *in cellulo* functional analysis indicate that only CHSY1 and CHSY3 are catalytically active. This conclusion is further supported by the architecture of the catalytic sites. Key

residues in the N-terminal domain (D261, D263, and H394 in CHSY3), which are critical for GlcA transfer, and in the C-terminal domain (D718, D720, and H831 in CHSY3), which are essential for GalNAc transfer, are highly conserved in CHSY1 and CHSY3 but are absent in CHPF and CHPF2 (Supplementary Fig. 18b)[32]. We speculate that only human CHSY1 and CHSY3 have retained their catalytic activities, while CHPF and CHPF2 have gradually lost their enzymatic function due to accumulated mutations over time. Nevertheless, CHPF and CHPF2 remain essential for complex formation and activity, likely due to their role in stabilizing the overall architecture of the CS polymerase complexes. Of note, the amino acid substitution D633N in CHSY1, previously identified in a patient with temtamy preaxial brachydactyly syndrome[44], was found to be critical for CS polymerization in our *in cellulo* studies. This residue, part of the DxD motif in the GalNAc-T domain, is essential for CS biosynthesis. This comparison underlines the relevance of the established *in cellulo* assay for studying the effects of specific mutations. On the other hand, to this date, only a few patient mutations have been identified in *CHSY1/CHSY3*[44,45] compared to the heparan sulfate polymerase EXT1/EXT2[29]. This is unlikely due to a lower importance of CS compared to HS, but rather to the redundancy within the system, as CHSY1 can compensate for the loss of CHSY3, and vice versa.

Our in vitro chain elongation studies suggest that all four purified CS polymerase complexes—CHSY1-CHPF, CHSY1-CHPF2, CHSY3-CHPF, and CHSY3-CHPF2—are capable of carrying out CS chain polymerization (Fig. 3b). One possible reason for the existence of multiple CS polymerase complexes could be the tissue-specific expression patterns of the CS biosynthetic genes. For example, *CHSY1* is more broadly expressed across tissues, whereas *CHSY3* appears to be predominantly expressed in neuronal and glial cells[31,46,47]. Another reason could be that different CS polymerase complexes produce CS chains of varying lengths, as previously suggested by Ogawa and co-workers[48]. At present, we cannot conclude whether the slight differences in CS polymerization activity observed among the complexes have biological relevance, as these variations could also be caused by differences in the thermal stability of the complexes or varying purification conditions (Fig. 2b). To further investigate potential differences in CS polymerase complex activity, additional in vitro kinetic studies using purified complexes, along with methodologies for precise chain-length determination, will be necessary. The presence of four different CS polymerase complexes in the cell could also allow for differential regulation, thereby fine-tuning chain length and sulfation motifs. Furthermore, these complexes may interact differently with downstream biosynthetic enzymes. For example, chondroitin 4-O-sulfotransferase-1, which introduces a sulfate group to position 4 of GalNAc residues, has been shown to regulate CS chain length[49]. The function of the middle domain remains elusive, but it may play a role in promoting protein–protein interactions with other cellular components, such as CS biosynthesis enzymes or the CSPG substrate. The contribution of the N-terminal anchoring helices to promoting oligomerization remains an important question. AlphaFold2 predictions of the full-length CS polymerase complexes suggest a potential coiled-coil formation between the N-terminal helices of CHSY3–CHSY1, CHSY3–CHPF, and CHSY3–CHPF2. Interestingly, the CHSY3–CHPF interface contains a glycophorin A-like motif (GxxxG), which is commonly observed in transmembrane helix interactions (Supplementary Fig. 19)[50]. All these questions warrant further investigation.

## Methods

### Structure prediction using AlphaFold 2
To study the formation of homo- and heterodimeric CS polymerase complexes, we performed model predictions for all combinations of the four human CS synthases using AlphaFold 2[33]. The protein sequences were retrieved from UniProt[20] and trimmed to remove the membrane-spanning helix. The resulting constructs—CHSY1 (Q86X52, aa68-aa802), CHPF (Q8IZ52, aa81-aa775), CHSY3 (Q70JA7, aa157-aa882), and CHPF2 (Q9P2E5, aa57-aa772)—were used as input sequences in ColabFold, which is based on AlphaFold 2 using the options num_models = 5 and num_cycle = 3[34,51–53]. The best-ranked model was used for subsequent analysis in ChimeraX[54]. To investigate potential interactions between the N-terminal anchoring helices, AlphaFold 2 predictions for the full-length CS synthases were performed.

### Co-expression study
To test soluble and secreted expression of human homo- and heterodimeric CS polymerase complexes, the coding regions of N-terminal truncated CHSY1 (aa68-aa802), CHPF (aa81-aa775), CHSY3 (aa157-aa882), and CHPF2 (aa57-aa772) were inserted into the pHR-CMV-TetO2 vector[55], utilizing the restriction sites AgeI and KpnI (Supplementary Table 10). The resulting constructs contain an N-terminal chicken RPTP sigma signal peptide, followed by a SUMO fusion protein, a 6x His tag, and a 3C protease cleavage site. CHSY1 and CHSY3 feature an Avi tag (biotin acceptor peptide) at their C-terminus, whereas CHPF and CHPF2 contain another 3C site followed by a twin streptavidin tag. 20 mL of human Freestyle 293-F cells (Thermo Fisher, R79007), cultured in FreeStyle™ 293 medium, were transfected or co-transfected using polyethylenimine (PEI25K, Polysciences, 23966-100) at a PEI to DNA mass ratio of 2:1 (single-transfection) or 2:0.5:0.5 (co-transfections). The cells were then incubated in a shaking incubator at 37 °C with 8% CO$_2$ for 96 h. Supernatants were collected by centrifugation at 500 x $g$ for 5 min at room temperature. Co-expression was further investigated in HeLa cells (ATCC, CCL-2™) and CHO K1 cells (kindly provided by Jeffrey D. Esko)[56]. HeLa cells were cultured in DMEM GlutaMAX™ supplemented with pyruvate and 10% fetal bovine serum (FBS), while CHO cells were maintained in DMEM/F-12 GlutaMAX™ with 10% FBS. For transfection, 2 mL of HeLa or CHO cells were transfected and co-transfected using jetPRIME® transfection reagent (Ozyme). Transient transfections were carried out at a jetPRIME®:DNA mass ratio of 3:1 for single-transfection, or 3:0.5:0.5 for co-transfections. Cells were then incubated at 37 °C with 8% CO$_2$ for 48 h. Supernatants were collected by centrifugation at 500 × $g$ for 5 min at room temperature. Transient expression was analyzed by western blot analysis using an anti-His HRP antibody (Sigma, A7058) at a dilution of 1:2000. Chemiluminescent signal was detected upon addition of Chemiluminescent HRP substrate (Sigma, A7058) on a ChemiDoc™ MP imaging system (Bio-Rad).

### Purification of CS polymerase complexes
600 mL of Freestyle HEK293-F cells (2 × 106 cells/mL) were co-transfected with the constructs and conditions described above. The cell supernatant was recovered by two consecutive centrifugation steps at 500 x $g$ for 5 min and 6000 x $g$ for 15 min, respectively. The supernatant was filtered through a 0.45 µM sterile filter (Merck), and buffer was added to a final concentration of 50 mM Tris pH 7.8, 150 mM NaCl, 10 mM imidazole, and 10 mM MgCl$_2$. All subsequent steps were performed at 4 °C. The supernatants were loaded onto a 5 mL HisTrap™ HP column (Cytiva). The column was washed with 5 column volumes of buffer A (50 mM Tris pH 7.8, 150 mM NaCl) containing 50 mM imidazole. The CS polymerase complexes were eluted with buffer A containing 250 mM imidazole. CS polymerase complexes were dialyzed using a dialysis membrane (Spectra/Por® MWCO 3.5 kD) in buffer B (50 mM Tris, pH 7.8, 500 mM NaCl) for 16 h at 4 °C in the presence of 500 µg 3C protease. A second HisTrap™ HP purification step was carried out, and protein complexes were eluted at an imidazole concentration of 100 mM. In this step, a 1 mL HisTrap™ HP column was coupled with two 5 mL HiTrap™ desalting columns for buffer exchange. CHSY1-CHPF and CHSY3-CHPF were stored in 20 mM MES at pH 6.5, 150 mM NaCl, while CHSY1-CHPF2 and CHSY3-CHPF2 were stored in 50 mM Tris at pH 7.8, 500 mM NaCl. Protein samples were

concentrated to 0.3–1 mg/mL and flash frozen in liquid nitrogen before storage at −80 °C for subsequent functional and structural analysis. Point mutations in CHSY3 constructs were introduced using site-directed mutagenesis PCR (Supplementary Table 11). Mutant CS polymerase complexes were purified using the same protocol as for the wild-type complexes.

## Nano-differential scanning fluorimetry

The thermal stability of purified CS polymerase wild-type and CHSY3 mutant-containing CHSY3-CHPF complexes was measured using nanoDSF. Protein complexes were diluted with 20 mM MES pH 6.5, 150 mM NaCl buffer to a final concentration of 0.2 mg/mL and loaded into glass capillaries (Nanotemper). Melting curves were recorded on a Prometheus NT48 device (Nanotemper) using an excitation power of 100% and a temperature gradient from 15 °C to 95 °C using a temperature slope of 1.5 °C per minute. Melting points of duplicate measurements were analyzed using the PR ThermControl v.2.3.1 software.

## Mass photometry analysis

Mass photometry analysis was performed using a Refeyn OneMP instrument. CS polymerase wild-type and CHSY3 mutant-containing CHSY3-CHPF complexes were diluted with a buffer containing 20 mM MES, pH 6.5, 150 mM NaCl to a final concentration of around 25 nM. Microscope coverslips (24 × 50 mm, 170 µm) were thoroughly cleaned with $H_2O$ and isopropanol before being air-dried. A reusable gasket (Grace Bio-Labs) was placed onto the clean coverslip, and 19 µL of the buffer used for protein dilution was applied. The focal position was identified automatically using Refeyn AcquireMP 2024.1.1 software, and then 1 µL of protein standard or diluted CS polymerase complex was added. Mass calibration was performed using a 1:20 diluted NativeMark™ unstained protein standard solution (Thermo Fisher Scientific). 60 s movies were recorded, corresponding to 6000 frames, using a regular field-of-view (18 µm²) acquisition setting. The corresponding masses for detected particles were calculated using Refeyn DiscoverMP 2024.1.0 software. Mass kernel density was estimated with an 8 kDa bandwidth. To investigate the behavior of the CS polymerase complex under denaturing conditions, 250 nM CHSY3−CHPF was incubated with 5.4 M urea at room temperature for 5 min. Both urea-treated and untreated CHSY3−CHPF samples were subsequently diluted in 20 mM MES pH 6.5, 150 mM NaCl to a final concentration of 12.5 nM and analyzed by mass photometry. Calibration was performed in the same conditions after incubation with 5.4 M urea.

## Chemo-enzymatic synthesis of penta- and hexasaccharide peptides

A peptide derived from colony-stimulating factor 1 (CSF1, EEASGEAS) was chemically synthesized with a TAMRA fluorescent label at the N-terminal position (SB-peptide). The pentasaccharide (GalNAc-GlcA-Gal-Gal-Xyl) addition was realised in a two-step process. The first step was to prepare the tetrasaccharide linker[14]. To do so, 300 µM CSF1 was incubated with 1 µM XylT1, 8.4 µM β4GalT7, 1 mM UDP-Xyl, and 1 mM UDP-Gal in a buffer containing 25 mM MES pH 6.5, 125 mM NaCl, 2.5 mM MgCl₂, 2.5 mM MnCl₂ with a final reaction volume of 3 mL. After 16 h incubation at room temperature, 2 µM β3GalT6, 2 µM GlcAT-1, 1 mM UDP-GlcA, and 1 mM UDP-Gal were added. After 24 h, the sample was loaded onto two Superdex Peptide 10/300 GL columns (Cytiva), connected in series and pre-equilibrated in 25 mM MES pH 6.5, 250 mM NaCl. Elution fractions were analyzed by 25% acrylamide gel electrophoresis using a fluorescence imaging system. The tetrasaccharide-CSF1 peptide-containing fractions were pooled and dialyzed against 4 L of dH₂O, lyophilized, and resuspended in 1 mL of dH₂O. In a second step, the fifth sugar, GalNAc, was enzymatically added using purified CHSY3-CHPF complex. A 2 mL reaction mixture containing 300 µM tetrasaccharide-CSF1 peptide, 1 mM UDP-GalNAc, and 1 µM CHSY3-CHPF was prepared in 50 mM MES pH 6.5, 50 mM

NaCl, 5 mM MnCl₂, and 2.5 mM MgCl₂. After 16 h incubation in the dark at 37 °C, the sample was purified and analyzed as described above. The concentration of Penta-CSF1 was determined using a titration curve with non-glycosylated peptides, loaded onto a 25% acrylamide gel, and by correlating the band intensities quantified using Image Lab v.6.1 (Bio-Rad). This Penta-CSF1 peptide served as a substrate for the synthesis of the Hexa-CSF1 peptide. A 200 µL reaction mixture was prepared containing 100 µM Penta-CSF1, 1 mM UDP-GlcA, 1 µM CHSY3-CHPF, 50 mM MES pH 6.5, 50 mM NaCl, 5 mM MnCl₂, and 2.5 mM MgCl₂. After 16 h incubation in the dark at 37 °C, the reaction products were purified and analyzed as described above. All glyco-peptides were stored at −20 °C.

## In vitro glycosylation assays

FACE assay using chemo-enzymatically synthesized fluorescent Penta-CSF1 and Hexa-CSF1 peptide substrates was used to follow in vitro CS chain elongation by purified CS polymerase complexes. CS chain elongation reaction mixture contained 3.5 µM Penta-CSF1 acceptor substrate, 3.5 mM UDP-GalNAc and 3.5 mM UDP-GlcA donor substrates, 0.35 µM wild-type or mutant CS polymerase complexes, and reaction buffer (50 mM MES pH 6.5, 50 mM NaCl, 5 mM MgCl₂, 5 mM MnCl₂). The final reaction volume was 400 µL. Reactions for wild-type complexes were incubated in the dark for 24 h at 30 °C, and mutant complexes for 4 h at 37 °C, before being stopped by heating the samples to 85 °C for 10 min. 1 µl reaction mix was diluted with 19 µl loading buffer containing 20% (v/v) glycerol and trace amounts of phenol red, from which 5 µl were loaded onto a 25% polyacrylamide gel. Bands were detected using a ChemiDoc™ MP imaging system (Bio-Rad) and the Alexa 546 nm filter. The bands were quantified using Image Lab 6.0.1.Cordeiro RL

For the CHSY3$^{mutant}$-CHPF$^{WT}$ complexes, each reaction contained 125 nM of enzyme complex, 1 mM UDP-GalNAc and 1 mM UDP-GlcA (as donor substrates), and 1 µM fluorescent Penta-CSF1 or 1 µM fluorescent Hexa-CSF1 (as acceptor substrates). All reactions were carried out in a total volume of 20 µL using a reaction buffer of 50 mM MES pH 6.5, 50 mM NaCl, 5 mM MgCl₂, and 5 mM MnCl₂. Reactions for wild-type complexes were incubated for 2 h at 30 °C, after which the reactions were stopped by heating to 85 °C for 10 min.

To perform chondroitinase digestion of the CHSY3-CHPF reaction product, a reaction mixture was prepared containing 1 µM fluorescent Penta-CSF1 as the acceptor substrate, 1 mM UDP GalNAc and 1 mM UDP GlcA as donor substrates, and 125 nM CHSY3-CHPF complex, 50 mM MES pH 6.5, 50 mM NaCl, 5 mM MgCl₂, and 5 mM MnCl₂, all in a total reaction volume of 40 µL. The mixture was incubated for 24 h at 37 °C in the dark. The sample was heated at 85 °C for 10 min to stop the reaction. For chondroitinase ABC digestion, 15 µL reaction mixture was added to 1 µL chondroitinase ABC (25 mU/µL, Sigma C3667), 100 mM sodium acetate, pH 7.4, and 0.5 mM CaCl₂. The mixture was incubated for 16 h at 37 °C, and the reaction was stopped by heating at 85 °C for 10 min.

To determine the importance of metal ions for GlcA and GalNAc transfer, similar reactions were prepared, including either 10 mM ethylenediaminetetraacetic acid, 10 mM MgCl₂, 10 mM MnCl₂, or 10 mM CaCl₂ or without any added cations. The reactions were incubated for 20 min. For the rescue experiment, CS polymerase complex concentrations varied between 62.5 nM and 125 nM. The reactions were incubated for 2 h.

## Size determination of CS chains by size-exclusion chromatography

In vitro glycosylation reactions from wild-type CS polymerase complexes were lyophilised using an Alpha 1-2 LD plus instrument (Martin Christ) and redissolved in 100 µL of dH₂O. Samples were then injected onto a Superdex 200 10/300 GL column (Cytiva), pre-equilibrated with 50 mM Tris pH 7.4, 100 mM NaCl. Detection was performed using the

520 nm channel. The V0 and the total bed volume of the column (Vt) were determined by injecting 200 μL of 2 mg/mL Dextran Blue and 500 mM imidazole (Sigma), respectively, using the same buffer as described above. Additionally, the column was calibrated by injecting 200 μL of 10 mg/mL FITC-Dextran markers with different MW: 4 kDa, 40 kDa, 70 kDa and 150 kDa (Sigma 68059, 53379, 53471, 74817). The corresponding elution volumes (Ve) were converted into Kav, using the formula Kav = (Ve−V0)/(Vt−V0). A calibration curve, Kav = f(log(MW)), was plotted and used to estimate the molecular weight of synthesized CS chains.

## Mass spectrometry analysis
We utilized a matrix-assisted laser desorption ionisation (MALDI)-time-of-flight (TOF)/TOF mass spectrometer (Autoflex MAX, Bruker Daltonics, Bremen, Germany). A Bruker proprietary (Nd:YAG) 355 nm smartbeamTM II solid-state laser (maximum frequency: 2 kHz) was used for ionisation. Each mass spectrum was generated by averaging 10,000 laser shots, randomly acquired from various target spots. Glycosylated peptides were analyzed in reflector positive ion mode, detecting the 800–3000 m/z range. Reaction mixtures containing peptides were diluted in 5% formic acid (FA) solution to obtain 5 μM samples. The procedure for depositing samples on the MALDI plate (Bruker AnchorChip™) was as follows: a thin layer composed of a saturated solution of α-cyano-4-hydroxycinnamic acid matrix in acetonitrile was prepared[57]. On the TL, 0.8 μL of the peptide sample was loaded. Then 0.8 μL of a mix of α-cyano-4-hydroxycinnamic acid and 2,5-dihydroxybenzoic acid matrices was put on each spot. The matrices were left to dry at room temperature. 0.6 μL of a standard calibrator (peptide standard II, Bruker Daltonics) was added next to the samples, as well as 0.6 μL of the matrix solution. Intact peptides spectra were processed using FlexAnalysis™ (version 3.4). The sophisticated detection algorithm of the Numerical Annotation Procedure allowed the identification of peaks with a signal-to-noise ratio threshold of 2 and a maximum number of 500 detected peaks.

## Cryo-EM sample preparation and data collection
Freshly purified CHSY3-CHPF complex, diluted in 20 mM MES pH 6.5, 100 mM NaCl, and 2.5 mM MnCl₂, was mixed with 10 mM UDP and 500 μM octasaccharide (GalNAc-GlcA)₄[58] and incubated for 1 h on ice. The final protein concentration was 0.6 mg/mL. 4 μL sample was applied onto a glow-discharged carbon-coated copper Quantifoil® R 1.2/1.3 300 Mesh grid, excess liquid was removed by blotting for 5 s, and the grid was plunge-frozen in liquid ethane cooled by liquid nitrogen using a Vitrobot Mark IV (FEI) plunge freezer, with the chamber temperature set to 4 °C and 100% humidity. Grids were screened using a 200 kV Glacios (FEI) electron microscope. Data collection was carried out at the European Synchrotron Radiation Facility (ESRF) on a 300 kV Titan Krios G4 (Thermo Fisher Scientific) transmission electron microscope (beamline CM02), equipped with a cold field emission gun, a Selectris X energy filter, and a Falcon 4i electron detector. 12,169 movies were collected with a total electron dose of 50 e⁻/Å² at a magnification of 165,000x and a raw pixel size of 0.73 Å.

## Cryo-EM image processing, structure building, and refinement
Cryo-EM data analysis was performed using the cryoSPARC v3.3.1 software. Movies were imported, and patch motion correction and patch CTF estimation were carried out. Upon manual inspection, 7598 micrographs were selected. An AlphaFold 2 predicted model of the CHSY3-CHPF complex was converted into a map using EMAN's pdb2mrc command[59], low-pass filtered to 8 Å, and used for several rounds of template-based particle picking, resulting in a total of 4,534,470 selected particles. Particles were extracted with a box size of 196 pixels and 2× binned. 2D classification was performed to remove junk particles, and particles were further sorted using several rounds of heterogeneous refinement. After removing duplicate particles, a final set of 166,015 particles was obtained, which was used for ab initio reconstruction and non-uniform refinement, resulting in a 3D reconstruction with a global resolution of 3.0 Å based on a FSC cutoff of 0.143. Local resolution-filtered full map was generated using a B-factor of −40. In addition, local refinements were performed for the N-terminal and C-terminal parts of the complex, resulting in maps with approximate nominal resolutions of 2.9 Å and 3.0 Å, respectively. These maps were also local resolution filtered using B-factors of −40. A composite map was generated by adding the N-terminal and C-terminal maps together using the vop maximum command in UCSF ChimeraX v.1.6.1[54]. The AlphaFold 2 predicted model of the CHSY3-CHPF complex was docked into the EM map using UCSF ChimeraX v.1.6.1[54]. The model was trimmed to remove the N-terminal and C-terminal amino acid residues as well as loops for which no EM density was observed. The model was then manually refined using WinCoot[60]. A UDP molecule, two Mn²⁺ ions, and three N-acetylglucosamines were placed into the corresponding EM density. Several rounds of real-space refinement in PHENIX v.1.20.1.4487 and v.1.21.1.5286[61] and manual building in WinCoot v.0.0.8.93 and v.0.9.8.7 were performed. The final model was validated using MolProbity v.4.5.2[62]. Figures were prepared using UCSF ChimeraX v.1.6.1[63].

## In cellulo complementation assay
A complementation assay was designed to test the ability of wild-type and mutant CHSY1 and CHSY3 proteins to carry out CS chain polymerization inside a cell. Synthetic genes encoding full-length CHSY1, CHSY3, CHPF, and CHPF2 were inserted into the pTWIST-CMV-Beta-Globin-WPRE-Neo vector (Twist Bioscience). The constructs also encoded an N-terminal 3x FLAG tag, an 8x His tag, and a 3C cleavage site. Point mutations were introduced through site-directed mutagenesis PCR (Supplementary Table 11). HEK293-6E wild-type and *CHSY1/CHSY3* double KO cells[23] were initially cultured as adherent cells in DMEM Gluta MAX™ medium, containing 20% FBS, 1% penicillin and streptomycin at 37 °C and 8% CO₂. The adherent cells were then detached and transferred into suspension culture using FreeStyle™ medium and placed into a shaking incubator set to 37 °C and 8% CO₂. Double KO cells (2 × 10⁶ cells/mL) were co-transfected with wild-type and mutant-containing CHSY1-CHPF and CHSY3-CHPF constructs using PEI (PEI:DNA₁:DNA₂ = 2:0.5:0.5). After 36 h, cells were harvested by centrifugation. To detect cell surface CS levels, 1 × 10⁶ cells were incubated with an anti-CS antibody (Sigma, C8035 clone 56, dilution 1:250) for 1 h on ice. Cells were washed twice with a PBS buffer containing 1% (w/v) bovine serum albumin (BSA) and then incubated with an Alexa Fluor 488-labeled anti-mouse secondary antibody (Jackson Immuno Research, 115-545-075) at a dilution of 1:300 for 1 h on ice. After three subsequent wash steps with the same buffer, the cells were fixed using 4% paraformaldehyde (PFA) for 15 min, again followed by three washing steps. Finally, cells were resuspended in 200 μL PBS buffer containing 1% (w/v) BSA and analyzed. To detect the transfection efficiency, 2 × 10⁶ cells were fixed with 4% PFA for 10 min at room temperature, washed with PBS, and permeabilized with 0.1% Triton X-100 for 10 min at room temperature. After washing with PBS buffer containing 1% (w/v) BSA, the cells were labeled with primary anti-FLAG antibody (Sigma, F1804) at a dilution of 1:300 for 1 h on ice, followed by two wash steps. Next, a cyanine3-conjugated anti-mouse antibody was added at a dilution of 1:300 (Jackson ImmunoResearch, 115-165-068) and cells incubated for 1 h on ice. Cells were finally washed, fixed with 4% PFA for 15 min on ice, and resuspended in 300 μL wash buffer. Flow cytometry experiments were performed on a Miltenyi Biotech flow cytometer and analyzed using the Macs Quant v.2.13.0 software (Miltenyi) on a population of approximately 30,000 intact cells. The transfection efficiency was quantified by dividing the number of cells successfully transfected by the total cell count. The applied gating strategy is exemplified in Supplementary Fig 12a. To quantify CS abundance, the median background signal (only anti-Alexa Fluor 488

antibody) was subtracted from the median signal obtained for staining with both anti-CS antibody and anti-Alexa Fluor 488 antibody. Median values were calculated for the full cell population after omitting a small population of dead cells. Resulting values were normalized to HEK293-6E WT cells (100%).

## CHPF-CHPF2 KO cell line generation

Adherent HEK293 (ATCC, CRL-1573) cells were maintained in 6-well plates at 37 °C and 5% CO2 in Dulbecco's Modified Eagle Medium F-12 Nutrient Mixture (DMEM/F-12; Gibco, 21331), supplemented with 10% FBS (Gibco, A5256801) and 1× GlutaMAX (Gibco, 35050-038). On the day of transfection, ~70% confluent cells were co-transfected with 1 μg of validated gRNA-encoding plasmid[64] and 1 μg of GFP-tagged Cas9-expressing plasmid (Addgene, 68371) by Lipofectamine 3000 (Invitrogen, L3000008) or FectoPRO (Polyplus, 101000014) according to the manufacturer's instructions. 24 h post-transfection, cells with GFP signals were enriched by fluorescence-activated cell sorting on a SH800 (Sony)[65]. 1–2 weeks later, the bulk-sorted pool was further single-cell-sorted into flat-bottom 96-well plates with DMEM/F-12, supplemented with 10% FBS, 1× GlutaMAX, 1% Pen Strep (Gibco, 15070-063, final concentration: 5 U/mL penicillin, 5 μg/mL streptomycin) to obtain single clones. 2 weeks later, clones with double KO editing patterns were screened by indel detection by amplicon analysis[66]. Sequences of double KO clones were resolved by Sanger sequencing (Eurofins) (Supplementary Table 9).

## Extraction and purification of GAGs from HEK293 cells

$1 \times 10^7$ cells (>95% viability) were washed in PBS (Sigma-Aldrich, D8537) three times and resuspended in 1 mL cell lysis buffer: 50 mM Tris-HCl (pH 7.3-7.6, Sigma-Aldrich, T5941), 10 mM $CaCl_2$ (Merck), 0.1% Triton X-100 (Sigma-Aldrich, T9284), and 1 mg/mL freshly prepared pronase (Roche, 10165921001). Cell lysis and protein digestion reactions were carried out overnight at 37 °C and were terminated by 98 °C heat inactivation for 10 min on the next day. Subsequently, final concentrations of 2 mM $MgCl_2$ (Sigma-Aldrich, M2670) and 10 μg/mL DNase I (Roche, 10104159001) were added, followed by incubation at 37 °C for 2 h. The DEAE beads were washed with 1.25 M NaCl and equilibrated with 20 mM sodium acetate (Merck) and 0.1 M NaCl (VWR Chemicals). Samples were then loaded onto equilibrated DEAE-Sepharose columns (Sigma-Aldrich, DFF100). The columns were previously equilibrated with 20 mM sodium acetate (Merck) and 0.1 M NaCl (VWR Chemicals), and purified GAGs were eluted with 1.25 M NaCl. GAGs from the eluate were precipitated by the addition of 3 volumes of cold sodium acetate-saturated 100% ethanol (VWR Chemicals) and incubated at −20 °C overnight. Pellets containing GAGs were collected by centrifugation at $21,000 \times g$ for 20 min at 4 °C and dried by SpeedVac (Eppendorf concentrator plus). Purified GAGs were reconstituted in 50 μL of Milli-Q water.

## Chondroitinase digestion and RP-UPLC disaccharide analysis of AMAC-labeled disaccharides

To digest purified CS chains into disaccharide units, purified GAG samples were added to a digestion buffer with final concentrations of 50 mM sodium acetate (pH = 7), 1 μM $CaCl_2$, and 0.25 mU/μL chondroitinase ABC (Sigma-Aldrich, C2905). The reaction was carried out at 37 °C overnight. Digested samples were dried by SpeedVac.

Dried chondroitinase digested samples were labeled with 2-aminoacridone (AMAC; MCE, HY-103594) by firstly resuspending samples in 5 μL of 0.1 M AMAC in 3:17 (v/v) acetic acid (VWR Chemicals)/dimethyl sulfoxide (Sigma-Aldrich), followed by an incubation at room temperature for 15 min. Then 5 μL of freshly prepared 1M sodium cyanoborohydride (Sigma-Aldrich, 156159) was added, and the reactions were incubated at 45 °C for 2h. The reaction products were dried by SpeedVac, and excess AMAC was removed by two rounds of resuspension in 500 μL of acetone (Merck), followed by a centrifugation step at $21,100 \times g$ for 20 min at 4 °C. AMAC-labeled pellet was obtained by centrifugation at $21,000 \times g$ for 20 min at 4 °C. AMAC-labeled samples were then dissolved in 2% acetonitrile (VWR Chemicals) and analyzed on a Waters Acquity RP-UPLC system equipped with a fluorescence detector with a BEH C18 column (2.1 × 150 mm, 1.7 μm; Waters), detecting the fluorescence signal at 525 nm. A standard mix of AMAC-labeled disaccharides (20 pmol of each) was analyzed immediately before the samples. Commercially available disaccharide standards were purchased from Iduron and Sigma-Aldrich.

## Reporting summary

Further information on research design is available in the Nature Portfolio Reporting Summary linked to this article.

## Data availability

The atomic coordinates for the CHSY3-CHPF complex have been deposited in the Protein Data Bank [https://www.rcsb.org/] with the PDB ID: 9Q8Z. The Cryo-EM maps have been deposited in the Electron Microscopy Data Bank under the following IDs: EMD-52913 (composite map), EMD-53018 (consensus map), EMD-53011 (focused refinement of the N-terminal part), and EMD-53012 (focused refinement of the C-terminal part). Mass spectrometry raw data is available through the Figshare repository [https://doi.org/10.6084/m9.figshare.28615220]. AlphaFold Predictions are available as Source Data. Unless otherwise stated, all data supporting the results of this study can be found in the article, supplementary, and source data files. Source data are provided with this paper.

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

## Acknowledgements

We thank Hugues Lortat-Jacob for his regular feedback on project ideas and his feedback on the manuscript. We thank Caroline Mas for assistance and access to mass photometry analysis, Michel Thepaut for access to the nano-differential scanning fluorimetry instrument, Jean-Philippe Kleman for help to acquire and analyze flow cytometry data, and Manal Hal Majed for performing site-directed mutagenesis PCR. We thank Shibom Basu and EMBL Grenoble for assistance and access to AlphaFold 2 predictions. We thank Eleftherios Zarkadas for support during cryoEM specimen optimization and for cryo-EM data acquisition, and Guy Schoehn for establishing and managing the IBS-ISBG cryo-EM platform and for providing access, training, and support. We thank Chrystel Lopin for providing the chemically synthesized octasaccharide compound and Rebecca Miller for sharing CHSY1/CHSY3 KO cell lines.

This work used the EM facility, the cell imaging platform, the mass spectrometry facility and the biophysics characterization platform of the Grenoble Instruct-ERIC Center (ISBG; UAR 3518 CNRS-CEA-UGA-EMBL) within the Grenoble Partnership for Structural Biology (PSB), supported by FRISBI (ANR-10-INBS-0005-02) and GRAL, financed within the University Grenoble Alpes graduate school (Ecoles Universitaires de Recherche) CBH-EUR-GS (ANR-17-EURE-0003). The IBS Electron Microscope facility is supported by the Auvergne-Rhône-Alpes Region, the Fonds Feder, the Fondation pour la Recherche Médicale and GIS-IbiSA. The cryo-EM data were acquired at the CM02 CRG beamline operated by IBS at the ESRF in Grenoble. The purchase of this microscope was funded by the EquipEx+ France Cryo-EM project (ANR-21-ESRE-0046). This work was funded by the Impulscience® Program of the Bettencourt Schueller Foundation, project name (GlycoPol, R.W.), by the ANR GlycoLink, grant number (ANR-23-CE44-0011, R.W.), a Labex-GRAL PhD fellowship, and the "Investissements d'avenir" program Glyco@Alps, grant number (ANR-15-IDEX-02, R.W.), a Novo Nordisk Foundation grant (NNF22OC0073736, R.L.M.), and the Copenhagen Center for Glycocalyx Research (DNRF196, R.L.M.).

## Author contributions

R.W. designed the project. P.D. performed AlphaFold 2 predictions. P.D. and M.W generated expression constructs. P.D. and M.B. performed co-expression tests and purified CS polymerase wild-type and mutant complexes. P.D. performed nanoDSF experiments. P.D. and S.D.V. performed mass photometry experiments. M.B. produced penta- and hexasaccharide peptides. P.D. and M.B. performed in vitro glycosylation assays. E.B.E. performed mass spectrometry analysis. M.M. and P.D. prepared and screened EM grids. R.W. and P.D. processed cryo-EM data. R.L.C. built the structure and carried out model refinement. P.D. R.L.C. and R.W. analyzed the structure. M.F. and P.D. performed the *in cellulo* complementation assay. H.S. M.N.N.G. and R.L.M. generated and analyzed CHPF/CHPF2 KO cells. P.D. and R.W. wrote the manuscript, and all authors contributed to its revision.

## Competing interests

The authors declare no competing interests.
