## [Transparent Peer Review file · Nature Communications]

Structural basis for human chondroitin sulfate chain polymerization

Corresponding Author: Dr Rebekka Wild

Version 0:

Reviewer comments:

Reviewer #1

(Remarks to the Author)

In this manuscript, the authors attempted to elucidate structural basis for human chondroitin sulfate (CS) chain polymerization. Via strategic experimental systems such as the efficient preparation methods for recombinant CS synthase complexes, cryo-electron microscopic (EM) analyses, and a novel in vitro glycosylation assay system, the authors confirmed four heterodimeric CS synthase complexes that have CS chain polymerization activity, and the functional role of the individual CS synthases: CHSY1 and CHSY3 as a bifunctional glycosyltransferase; CHPF and CHPF2 as a non-catalytic component of CS synthase complexes. In addition, based on the cryo-EM structure of CHSY3-CHPF complex and the rescue experiment by mixing the N-terminal CHSY3 mutant/CHPF and C-terminal CHSY3 mutant/CHPF, the authors suggested that CS chain polymerization follows a non-processive, disruptive mechanism. Experiments are carefully performed and all data presented in this manuscript is convincing. However, this study is largely overlapped previous studies such as Izumikawa et al. (2007) and Izumikawa et al. (2008). Therefore, this reviewer feels that the manuscript requires major revisions to ensure its novelty.

Major points #1

As shown in Fig. 1, the authors only detected the protein expression of four types of CS synthase complexes, and found that the other two complexes, CHSY1/CHSY3 and CHPF/CHPF2, and each CS synthase alone were not expressed at detectable levels. These results are inconsistent with previous reports and seem implausible. Are these trends specific to the expression system used, or are they general regardless cell types? It should also be confirmed whether the proteins whose expression could not be observed are due to their instability or whether they simply could not be secreted.

Major points #2

In the in vitro glycosylation assay system, how did the authors confirm whether the reaction products contain a polymerized chondroitin backbone? At least chondroitinase ABC treatment of the reaction products and their band shifts on the FACE analysis should be shown. In addition, to facilitate the evaluation of CS chain polymerization efficacy of the individual CS synthase complexes, the authors should show polymer size information on the FACE data. An additional gel filtration analysis of the reaction products may clarify this point.

Major points #3

In discussion, the authors concluded that CHPF and CHPF2 have a structural role that are required for enzymatic activity of CHSY1 and CHSY3. However, their involvement in cellular CS biosynthesis was not examined. Evaluation of CS abundance in CHPF/CHPF2 double KO cells should provide important insight into in cellulo functions of CHPF/CHPF2. If the level of CS abundance in the CHPF/CHPF2 double KO cells is almost equivalent to that in the CHSY1/CHSY3 double KO cells, the validity of the authors conclusions about the principal roles of four types of CS synthase complexes, CHSY1/CHPF, CHSY1/CHPF2, CHSY3/CHPF, and CHSY3/CHPF2, would be strengthened. By contrast, if the relatively higher level of CS abundance is observed in the CHPF/CHPF2 double KO cells, compared with that in the CHSY1/CHSY3 double KO cells, this suggests the functional involvement of CHSY1/CHSY3 complex in in cellulo CS biosynthesis.

Reviewer #2

(Remarks to the Author)

Dutta et al., in their manuscript "Structural basis for human chondroitin sulfate chain polymerization," have discovered and

characterized four heterodimeric complexes responsible for chondroitin sulfate chain polymerization in humans: CHSY1-CHPF, CHSY1-CHPF2, CHSY3-CHPF, and CHSY3-CHPF2. They demonstrated the activity of these complexes in vitro by developing a glycosylation assay based on chemoenzymatically synthesized fluorescent substrates. Additionally, they solved the cryo-EM structure of the CHSY3-CHPF complex. I find the manuscript interesting, easy to follow, and the findings valuable to the field.

I will focus on the molecular biophysics section, as assigned by the editor, specifically, the nDSF and Mass Photometry experiments.

nDSF:

In their differential scanning fluorimetry experiments, the authors show that all four heterocomplexes are folded, display cooperative unfolding, and exhibit similar T_m values. The experiments are well performed; however, the presentation of the results can be improved.

- I suggest the authors combine the replicate measurements and provide the T_m values with their standard deviations in brackets in Figure 2B.
- The figure legend should explain that the graph corresponds to the first derivative plot of the fluorescence 350/330 ratio.
- The fluorescence curves should be shown as a panel in Supplementary Figure 2.
- The authors also performed nDSF experiments on mutant complexes as controls, demonstrating that these are properly folded and exhibit similar T_m values as the wild-type complexes. The figure legend should be improved accordingly regarding the plotting of the First derivative.
- Supplementary Figure S9 should display the T_m values rounded to one decimal place, with standard deviation.

Mass Photometry:

The data analysis and representation for the Mass Photometry experiments need significant improvement.

- Please explain in the figure legends the expected molecular mass for each heterocomplex shown in Figure 2, Supplementary Figure S2, and Supplementary Figure S9.
- In the main text, comment on the discrepancies between expected and observed masses. It is common for mass photometry experiments to show shifts of approximately 20–30 kDa.
- One way to account for this is to provide an error interval corresponding to the standard deviation (SD) of the fitted Gaussians. The fitted means and standard deviations can be displayed, along with the number of counts (and percentage of total) under each fitted Gaussian distribution. If you would choose to fit the Gaussians. This method is in my opinion the best analysis (there are tools online to do this). Alternatively you could for each species (MP peak), write in the text the mass plus/minus standard error, where the standard error is calculated from the MP replicates.
- There are peaks around 75 kDa (in Figure 2 and S2) and over 300 kDa. The 75 kDa peak may correspond to a minor population of monomers or other impurities (I kind of discard this second since the SDS-PAGE appears quite clean). The >300 kDa species likely correspond to tetramers.

To test this, I suggest the authors repeat the MP measurements for one heterocomplex as example after incubating the sample in 5 M guanidinium HCl or 6 M urea for three hours. Use the same calibration curve as in the routine (NativeMark™). This treatment should result in the disappearance of the peak around 170 kDa and a shift toward the monomeric species.

Reviewer #3

(Remarks to the Author)

The manuscript by Poushalee Dutta et al. describes functional and structural analyses of enzymes involved in chondroitin (sulfate) biosynthesis. Chondroitin sulfate (CS) is an important extracellular matrix component in vertebrates, involved in many physiological processes. Understanding how this negatively charged polysaccharide is synthesized and secreted remains an important question.

Here, the authors analyze four components (CHSY1, CHSY3, CHPF, and CHPF2) implicated in CS biosynthesis using computational, biochemical, and structural approaches. They show that AlphaFold predicts stable heterodimeric complexes of the catalytically active subunits CHSY1 and CHSY3 with CHPF and CHPF2, determine the cryo-EM structure of the CHSY3-CHPF heterodimeric complex, and perform mutagenesis analysis of the CHSY3-CHPF complex in vitro and in a cell-based system.

Previous studies have already shown, albeit with some disagreement in the details, that heterodimerization is necessary for protein expression and chondroitin polymerization. However, a structure has not yet been provided, and the GalNAc and GlcA transferase activity assignments of the domains have not been biochemically confirmed.

Overall, considering existing knowledge in the field (which is inadequately referenced), the limited insights gained from their structural studies, and incomplete biochemical analyses, the novelty of the presented data is moderate.

Major points

Introduction:

The manuscript would benefit from an expanded and more balanced introduction section. This would not only demonstrate awareness and appreciation for the work of previous investigators, but also allow the readers (and reviewers) to better understand the context and novelty of the advance. In general, the manuscript is structured in a way that could mislead the reader into overestimating the novelty of the work presented.

In the introduction, it would be helpful to state:

1. CS is not just modified by sulfation; a significant amount is epimerized to form dermatan sulfate (Mikami & Kitagawa 2013, Ricard-Blum et al 2024).

2. Six, not four genes are association with CS chain polymerization (Mikami & Kitagawa 2013). For some reason ChGn-1 and ChGn-2 are never even mentioned in the manuscript. A comparison of the GT7 domains of the CHSY3 structure and ChGn AlphaFold structures would also be beneficial later in the text.
3. The first GalNAc is likely added by ChGn-1 and 2, which have only a GT7 domain.
4. ChGn-1/2 compete with EXTL3 for the linkage tetrasaccharide, and there are functionally important differences between heparan and chondroitin sulfate.
5. CHSY/CHPF enzymes are predicted to have an N-terminal GT31 domain and a C-terminal GT7 domain (CAZy).
6. Yada et al (2003) already showed that CHSY3 prefers Mn over Mg as a cofactor.
7. The GT31 domain is assumed to harbor β -1,3-GlcA-T activity, whereas the GT7 domain is assumed to harbor β -1,4-GalNAc-T activity (e.g. Filipek-Górniok et al. 2013). This is based on the activity of ChGn-1/2 being GalNAc-T but was also originally proposed based on the types of linkage generally formed by the two families (Gotoh et al 2002, ' β -glycosyltransferase motif' and ' β 4-glycosyltransferase motif').
8. Izumikawa et al. (2007) were apparently able to pull down CHSY3 using CHSY1 or CHPF, and CHSY1 using CHPF (note that CHSY3 = 'ChSy-2'). Izumikawa et al. (2008) were apparently able to pull down CHSY1 and CHPF using CHPF2, and CHPF2 using CHSY3 (note that CHPF2 = 'ChSy-3').
9. The currently accepted idea seems to be that CHSY1/3 forms a heterodimer with CHPF(2) (e.g. Petit et al 2021 <https://doi.org/10.1093/glycob/cwaa086>)
10. Sammon et al (2023) already found that 'CS polymerase subunits could not be expressed alone; the only combination that produced sufficient soluble protein for enzymological analysis was CHSY3/CHPF.'
11. Sammon et al (2023) appear to have already developed an assay for monitoring the activity of purified CHSY3/CHPF on a similar substrate as the one used here.

Results:

Fig. 1:

- Fig. 1a: Please state how this experiment was performed. Is this a Western blot of the culture supernatant or material bound to an affinity column? Is it possible that, when expressed individually, some proteins exhibit secretion defects and aggregate intracellularly?

- Fig. 1b: The AlphaFold predicted heterodimers lack their N-terminal transmembrane anchors. Since some of these anchors are predicted to contain glycoporphin A-like dimerization motifs, how could one exclude that dimerization of the TM helices could give rise to another level of oligomerization? Could this be tested experimentally?

Fig. 2c and similar FACE results throughout:

- The gels require a molecular weight marker to indicate mass changes and to allow cross comparisons with gels shown in different figures. Further, the gels are shown at an inappropriate contrast that hides weaker bands. These are visible from the source data after adjusting the contrast. The presence of impurities in the synthesized substrate should be acknowledged and considered when interpreting the potential products of in vitro synthesis reactions.

Fig. 3 and CHSY3-CHPF structure:

- For the resolution estimate of 3A it should be stated that this is only a nominal resolution estimate. The map quality of the N-terminal domain is closer to 3.5 to 4A over large regions.

- The authors do not discuss at all the architecture and potential function of the middle domain. Are there any structural homologs with known function?

- On several occasions, the authors refer to UDP as a donor substrate. UDP is not a substrate but a product of the enzymatic reaction (and perhaps an inhibitor). This should be distinguished. Further, the authors do not discuss any potential donor or acceptor binding sites (for example for GalNAc, GlcA, or the oligosaccharide). A potential oligosaccharide acceptor was added to the cryo EM mix but it is not stated whether it was resolved in the EM map.

- Lines 203: H394 is listed among residues that contact UDP, but it seems to be involved in cofactor (Mn²⁺) coordination.

- In general, the novelty of the identified residues is overstated since most of them belong to conserved motifs that have been well characterized in other enzymes.

- Fig. 4a: Including a sequence alignment of, for example, CHSY3 and CHPF would be helpful to illustrate catalytically important residues that are absent in CHPF/CHPF2.

- Fig. 4c and d: Molecular weight markers are required, and the images should be presented at an appropriate contrast level. Further, since both substrates were added to the reaction mix, it is important to characterize the products obtained, perhaps by MS/MS. Otherwise, it is difficult to determine what products have been formed, in particular considering the impurities present in the synthesized acceptor peptide.

- Fig. 4e: It should be acknowledged that the preference for Mn²⁺ has already been established. For this comparison, it would be informative to include a control in which no additional cation and no EDTA was added. This would indicate whether any cations copurify with the enzyme.

- Another disease related mutation of CHSY3 (P626R) could be added to the analysis.

Minor points:

- Introduction: For a general reader, please explain what is meant by transferase-II. What does 'II' refer to?

- Fig. 2b and corresponding text (lines 121-124): Annotating a single inflection point for the melting of the entire complex is misleading, in fact, if CHSY1-CHPF complexation turns out to be weaker, the negative inflection point seen around 50-55°C might be the actual quaternary structure melting point instead. The authors may simply state that the overall melting profile seem shifted towards a higher temperature in case of CHSY3-CHPF complex.

- Please avoid using 'electron' density when referring to EM map.

- The description of UDP/Mn binding to the GT31 domain in lines 171-172 is repeated almost immediately in lines 199-200

- Line 201: The authors should specify that the Dx/D motif is a common motif of GT-A enzymes, not all GTs. Further,

reference 36 seems inappropriate here. A better reference would be Wiggins & Munro 1998 (<https://doi.org/10.1073/pnas.95.14.7945>).

- Line 208 '4- β -N' should presumably be ' β -1,4-N'

- According to the glycobiology nomenclature, glycans should be represented with their non-reducing end to the left and reducing end to the right. The authors may want to consider this when representing the glycopeptide sequences in the various figure panels.

- CHSY3-CHPF glycosylation sites: Some glycosylation sites reveal fairly strong densities beyond the modeled GlcNAc sugar. It would be informative to include an additional supplemental figure that shows these densities. Also, there may be an additional glycosylated residue in CHPF (N361).

- Line 428: It is suggested that all *four* CS synthases might have had activity early in evolution. Firstly, the divergences of CHSY1 from CHSY3 and CHPF from CHPF2 only appear to have occurred in the vertebrate lineage (Petit et al 2021 <https://doi.org/10.1093/glycob/cwaa086>). A single CHSY and CHPF descendant is found in e.g. the sponge Amphimedon. It would be relatively easy to test the hypothesis above by analyzing their sequences as well as those in other distantly related orthologues.

- Is there any information on the tissue specific expression levels of CHSY1/3/CHPF/2?

- Supplementary Fig. 4: Can the molecular weights in panels b and c be indicated above each peak? It is important to show the individually observed masses in each spectrum so that the assignments can be verified in a case-by-case basis.

- Supplementary Fig. 12: Please indicate what total cell number was used to normalize the data shown in panels b and c. Was this the area boxed in green and red on the right side of panel a?

- Line 486: in which medium were the cells cultured? This is important for reproducibility as some media contain EDTA (which could interfere with nickel affinity chromatography).

- Typos: Line 61 'trough', line 547 'serie', and line 580 'spectromety'

- Line 25 and elsewhere: 'Distributive' and not 'disruptive' mechanism.

Version 1:

Reviewer comments:

Reviewer #1

(Remarks to the Author)

The authors have adequately addressed all my concerns. I have no further comments.

Reviewer #2

(Remarks to the Author)

The authors have addressed all my concerns regarding nDSF and Mass Photometry, and the molecular biophysics section is improved in the revised version.

Reviewer #3

(Remarks to the Author)

The revised manuscript by Dutta et al. has improved significantly. The authors have addressed most of our concerns. However, one issue remains that should be addressed, either experimentally or as an additional discussion.

In the previous review, we suggested considering dimerization of the transmembrane helices via the glycoporphin A motif as an additional or alternative mode of complex formation. The authors state in the rebuttal that AlphaFold predictions resulted in very low confidence in dimerization of this motif and thus dismiss the idea. It seems, however, that the opposite is the case.

First, the provided PAE plot for the CHPF/CHSY3 interaction shows a small area of high confidence (blue to white) for the transmembrane region.

Second, AlphaFold predicts the transmembrane helix dimerization with high confidence when using truncated CHSY3 and CHPF constructs. In fact, this is the only region that is predicted with high confidence in C-terminally truncated constructs.

Third, as found by a literature search, the transmembrane helix of CHPF2 (Uniprot: Q9P2E5) has previously been predicted by the CATM algorithm to form the strongest glycine zipper interaction of any protein in the entire human protein (Anderson et al 2017, JACS <https://doi.org/10.1021/jacs.7b07505>)

Considering these points, we believe that the possibility of homo- and/or heterodimerization mediated by the glycoporphin A motifs should be experimentally tested (ideally) or at least discussed.

Additional minor points:

Line 59: '4- β -N' should be ' β -1,4-N'

Fig. 3c: which combination of enzymes produced the chondroitin product used to test for chondroitinase sensitivity?

Page 9: the ordering and correspondence of supplementary figures to their references seems to be messed up here

Line 222: typo 'PSIA' (should be 'PISA')

Line: 265: Can M178 really be said to interact with the UDP uracil moiety? It is 6.2 Å from the uracil and 5.5 Å from the ribose.

Version 2:

Reviewer comments:

Reviewer #3

(Remarks to the Author)

The authors have addressed our remaining concerns. We don't have any additional comments.

Reviewer #1 (Remarks to the Author):

In this manuscript, the authors attempted to elucidate structural basis for human chondroitin sulfate (CS) chain polymerization. Via strategic experimental systems such as the efficient preparation methods for recombinant CS synthase complexes, cryo-electron microscopic (EM) analyses, and a novel in vitro glycosylation assay system, the authors confirmed four heterodimeric CS synthase complexes that have CS chain polymerization activity, and the functional role of the individual CS synthases: CHSY1 and CHSY3 as a bifunctional glycosyltransferase; CHPF and CHPF2 as a non-catalytic component of CS synthase complexes. In addition, based on the cryo-EM structure of CHSY3-CHPF complex and the rescue experiment by mixing the N-terminal CHSY3 mutant/CHPF and C-terminal CHSY3 mutant/CHPF, the authors suggested that CS chain polymerization follows a non-processive, disruptive mechanism.

Experiments are carefully performed and all data presented in this manuscript is convincing. However, this study is largely overlapped previous studies such as Izumikawa et al. (2007) and Izumikawa et al. (2008). Therefore, this reviewer feels that the manuscript requires major revisions to ensure its novelty.

Major points #1

As shown in Fig. 1, the authors only detected the protein expression of four types of CS synthase complexes, and found that the other two complexes, CHSY1/CHSY3 and CHPF/CHPF2, and each CS synthase alone were not expressed at detectable levels. These results are inconsistent with previous reports and seem implausible. Are these trends specific to the expression system used, or are they general regardless cell types? It should also be confirmed whether the proteins whose expression could not be observed are due to their instability or whether they simply could not be secreted.

Our answer:

To rule out that our observations were cell type specific, we performed addition co-expression studies in human HeLa and Chinese hamster ovary (CHO) cells. The following sentence and Supplementary Figure 1 was added to the manuscript:

"Additional co-expression experiments in HeLa cells confirmed that the formation of the four heterodimeric complexes is not cell-type specific (Supplementary Fig. 1c). Similar results were obtained in Chinese hamster ovary (CHO) cells, although protein expression in this system was near the detection limit (Supplementary Fig. 1d)."

To minimize the detection of misfolded or unfolded proteins retained within cells, we strategically chose a secreted expression system. This approach incorporates a native, cellular quality control mechanism, as only properly folded proteins are typically secreted. Nevertheless, to further investigate why certain proteins were not detected in the medium during our co-expression assays, we performed Western blot analysis on corresponding cell pellets. The results are presented in an additional paragraph in the manuscript and in the newly added Supplementary Fig. 1.

"We further investigated intracellular protein expression to determine whether the absence of secreted protein was due to a lack of expression or inefficient secretion. In addition to the correctly secreted heterodimeric complexes (CHSY1-CHPF, CHSY1-CHPF2, CHSY3-CHPF, and CHSY3-CHPF2), we observed bands indicative of intracellular CHPF and CHPF2 expression (Supplementary Fig. 1a, b). A possible explanation for the intracellular retention of CHPF/CHPF2 when expressed individually is that they either fail

to pass endoplasmic reticulum (ER) quality control or form heterodimeric complexes with endogenous, Golgi-localized CHSY1 or CHSY3."

We would like to emphasize that previous studies by Izukama investigated complex formation by overexpressing various combinations of proteins and assessing enzymatic activity in the cell supernatant, rather than using purified proteins. Therefore, these experiments do not directly demonstrate heterodimeric complex formation between the overexpressed proteins. It is also possible that the observed activity resulted from interactions between overexpressed and endogenous cellular proteins. Addressing this would have required the use of knockout (KO) cell lines. Moreover, pull-down experiments alone do not constitute definitive evidence of direct physical interaction between proteins. In contrast, our in cellulo data using CHSY1/CHSY3 double KO and CHPF/CHPF2 double KO cell lines strongly argue against the formation of catalytically active CHSY1–CHSY3 or CHPF–CHPF2 complexes (see major point #3).

Major points #2

In the in vitro glycosylation assay system, how did the authors confirm whether the reaction products contain a polymerized chondroitin backbone? At least chondroitinase ABC treatment of the reaction products and their band shifts on the FACE analysis should be shown. In addition, to facilitate the evaluation of CS chain polymerization efficacy of the individual CS synthase complexes, the authors should show polymer size information on the FACE data. An additional gel filtration analysis of the reaction products may clarify this point.

Our answer: We thank reviewer 1 for these suggestions. To provide additional proof that the reaction product observed during FACE analysis is indeed a polymerized chondroitin backbone, we performed chondroitinase ABC treatment. Figure 3c has been added to the manuscript along with the statement:

"Treatment of the reaction product with chondroitinase ABC confirmed that the observed band shift was due to chondroitin backbone polymerization (Fig. 3c)."

To study the size of generated polysaccharide chains, we performed size-exclusion chromatography experiments, as suggested by the reviewer. The following paragraph and a new Supplementary Figure 6 was added to the manuscript.

"To estimate the length of the synthesized chondroitin chains, reaction products were analyzed by size-exclusion chromatography and compared to commercial fluorescein isothiocyanate (FITC)-labelled dextran standards. Chondroitin generated by the CHSY3-CHPF2 showed an approximate molecular weight of 43 kDa. The reaction products of the other CS polymerase complexes were estimated to have molecular weights of approximately 137-151kDa. However, since they eluted near the void volume, their actual size might be larger (Supplementary Fig. 6). These molecular weights correspond to chains of 800 or more glycan units, suggesting that - except for CHSY3-CHPF2 - the CS polymerase complexes can efficiently elongate CS chains in vitro until nearly all substrates are consumed. "

Major points #3

In discussion, the authors concluded that CHPF and CHPF2 have a structural role that are required for enzymatic activity of CHSY1 and CHSY3. However, their involvement in cellular CS biosynthesis was not examined. Evaluation of CS abundance in CHPF/CHPF2 double KO cells should provide important insight into in cellulose functions of CHPF/CHPF2. If the level of CS abundance in the CHPF/CHPF2 double KO cells is almost equivalent to that in the CHSY1/CHSY3 double KO cells, the validity of the authors conclusions about the principal roles of four types of CS synthase complexes, CHSY1/CHPF, CHSY1/CHPF2, CHSY3/CHPF, and CHSY3/CHPF2, would be strengthened. By contrast, if the relatively higher level of CS abundance is observed in the CHPF/CHPF2 double KO cells, compared with that in the CHSY1/CHSY3 double KO cells, this suggests the functional involvement of CHSY1/CHSY3 complex in cellulose CS biosynthesis.

Our response: The analysis of a CHPF/CHPF2 double knockout cell line indeed provides valuable additional insight into the role of polymerization factors in heterodimeric complex formation. To address this point, we established a collaboration with the group of Dr. Rebecca L. Miller (University of Copenhagen), who conducted the additional experiments. The following paragraph has been added to the manuscript, along with a new Supplementary Figure 17 and the corresponding method sections.

"Our co-expression studies demonstrated that the presence of CHPF or CHPF2 is required for the production of secreted CS polymerase complexes (Fig. 1a). However, CHPF and CHPF2 alone are not sufficient to mediate CS chain elongation, as evidenced by the absence of CS chains in complexes containing mutant CHSY1 or CHSY3 (Fig. 6b,c). To further assess the role of CHPF and CHPF2 in CS biosynthesis, CHPF/CHPF2 double knockout cell lines were generated using CRISPR-Cas technology. Analysis of cell surface CS content revealed an absence of detectable CS (Supplementary Fig. 17, Supplementary Table 9), supporting the hypothesis that the presence of either CHPF or CHPF2 is essential for the formation of stable and functional CS polymerase complexes. These findings also argue against the formation of a functional CHSY1–CHSY3 complex in the absence of CHPF or CHPF2."

Reviewer #2 (Remarks to the Author):

Dutta et al., in their manuscript "Structural basis for human chondroitin sulfate chain polymerization," have discovered and characterized four heterodimeric complexes responsible for chondroitin sulfate chain polymerization in humans: CHSY1-CHPF, CHSY1-CHPF2, CHSY3-CHPF, and CHSY3-CHPF2. They demonstrated the activity of these complexes in vitro by developing a glycosylation assay based on chemoenzymatically synthesized fluorescent substrates. Additionally, they solved the cryo-EM structure of the CHSY3-CHPF complex. I find the manuscript interesting, easy to follow, and the findings valuable to the field.

I will focus on the molecular biophysics section, as assigned by the editor, specifically, the nDSF and Mass Photometry experiments.

nDSF:

In their differential scanning fluorimetry experiments, the authors show that all four heterocomplexes are folded, display cooperative unfolding, and exhibit similar T_m values. The experiments are well performed; however, the presentation of the results can be improved.

- I suggest the authors combine the replicate measurements and provide the T_m values with their standard deviations in brackets in Figure 2B.

Our answer: Ok, Figure 2b was updated accordingly.

- The figure legend should explain that the graph corresponds to the first derivative plot of the fluorescence 350/330 ratio.

Our answer: Done as suggested.

- The fluorescence curves should be shown as a panel in Supplementary Figure 2.

Our answer: Ok, the fluorescence traces were added in Supplementary Figure 3a.

- The authors also performed nDSF experiments on mutant complexes as controls, demonstrating that these are properly folded and exhibit similar T_m values as the wild-type complexes. The figure legend should be improved accordingly regarding the plotting of the First derivative.

Our answer: Done as suggested.

- Supplementary Figure S9 should display the T_m values rounded to one decimal place, with standard deviation.

Our answer: Supplementary Figure 12b and c were modified as suggested.

Mass Photometry:

The data analysis and representation for the Mass Photometry experiments need significant improvement.

- Please explain in the figure legends the expected molecular mass for each heterocomplex shown in Figure 2, Supplementary Figure S2, and Supplementary Figure S9.

Our answer: Done as suggested.

- In the main text, comment on the discrepancies between expected and observed masses. It is common for mass photometry experiments to show shifts of approximately 20–30 kDa.

Our answer: The following statement has been added to the article: "*Mass differences of up to 11 kDa between measured and theoretically calculated molecular weights can be attributed to N-linked glycosylation, as well as to the precision limits of this technique.*"

- One way to account for this is to provide an error interval corresponding to the standard deviation (SD) of the fitted gaussians. The fitted means and standard deviations can be displayed, along with the number of counts (and percentage of total) under each fitted Gaussian distribution. If you would choose to fit the Gaussians. This method is in my opinion the best analysis (there are tools online to do this). Alternatively you could for each

species (MP peak), write in the text the mass plus/minus standard error, where the standard error is calculated from the MP replicates.

Our answer: To provide this additional information, we have added Supplementary Table 2 and Supplementary Table 8.

- There are peaks around 75 kDa (in Figure 2 and S2) and over 300 kDa. The 75 kDa peak may correspond to a minor population of monomers or other impurities (I kind of discard this second since the SDS-PAGE appears quite clean). The >300 kDa species likely correspond to tetramers.

To test this, I suggest the authors repeat the MP measurements for one heterocomplex as example after incubating the sample in 5 M guanidinium HCl or 6 M urea for three hours. Use the same calibration curve as in the routine (NativeMark™). This treatment should result in the disappearance of the peak around 170 kDa and a shift toward the monomeric species.

Our answer: We thank reviewer 2 for suggesting this experiment to study the behavior of the CHSY3-CHPF complex under denaturing conditions. We have added the following sentence in the manuscript:

"Further mass photometry experiments demonstrated that the heterodimeric CHSY3-CHPF complex dissociates into monomeric proteins under denaturing conditions (Supplementary Figure 3b, Supplementary Table 3)."

Of note, we did not observe any peaks indicative of monomers or tetramers in the untreated sample this time (same batch of frozen protein as for previous experiments).

Reviewer #3 (Remarks to the Author):

The manuscript by Poushalee Dutta et al. describes functional and structural analyses of enzymes involved in chondroitin (sulfate) biosynthesis. Chondroitin sulfate (CS) is an important extracellular matrix component in vertebrates, involved in many physiological processes. Understanding how this negatively charged polysaccharide is synthesized and secreted remains an important question.

Here, the authors analyze four components (CHSY1, CHSY3, CHPF, and CHPF2) implicated in CS biosynthesis using computational, biochemical, and structural approaches. They show that AlphaFold predicts stable heterodimeric complexes of the catalytically active subunits CHSY1 and CHSY3 with CHPF and CHPF2, determine the cryo-EM structure of the CHSY3-CHPF heterodimeric complex, and perform mutagenesis analysis of the CHSY3-CHPF complex in vitro and in a cell-based system.

Previous studies have already shown, albeit with some disagreement in the details, that heterodimerization is necessary for protein expression and chondroitin polymerization. However, a structure has not yet been provided, and the GalNAc and GlcA transferase activity assignments of the domains have not been biochemically confirmed. Overall, considering existing knowledge in the field (which is inadequately referenced), the limited insights gained from their structural studies, and incomplete biochemical analyses, the novelty of the presented data is moderate.

Major points

Introduction:

The manuscript would benefit from an expanded and more balanced introduction section. This would not only demonstrate awareness and appreciation for the work of previous investigators, but also allow the readers (and reviewers) to better understand the context and novelty of the advance. In general, the manuscript is structured in a way that could mislead the reader into overestimating the novelty of the work presented.

In the introduction, it would be helpful to state:

1. CS is not just modified by sulfation; a significant amount is epimerized to form dermatan sulfate (Mikami & Kitagawa 2013, Ricard-Blum et al 2024).
2. Six, not four genes are association with CS chain polymerization (Mikami & Kitagawa 2013). For some reason ChGn-1 and ChGn-2 are never even mentioned in the manuscript. A comparison of the GT7 domains of the CHSY3 structure and ChGn AlphaFold structures would also be beneficial later in the text.
3. The first GalNAc is likely added by ChGn-1 and 2, which have only a GT7 domain.
4. ChGn-1/2 compete with EXTL3 for the linkage tetrasaccharide, and there are functionally important differences between heparan and chondroitin sulfate.
5. CHSY/CHPF enzymes are predicted to have an N-terminal GT31 domain and a C-terminal GT7 domain (CAZy).
6. Yada et al (2003) already showed that CHSY3 prefers Mn over Mg as a cofactor.
7. The GT31 domain is assumed to harbor β -1,3-GlcA-T activity, whereas the GT7 domain is assumed to harbor β -1,4-GalNAc-T activity (e.g. Filipek-Górniok et al. 2013). This is based on the activity of ChGn-1/2 being GalNAc-T but was also originally proposed based on the types of linkage generally formed by the two families (Gotoh et al 2002, ' β 3-glycosyltransferase motif' and ' β 4-glycosyltransferase motif').
8. Izumikawa et al. (2007) were apparently able to pull down CHSY3 using CHSY1 or CHPF, and CHSY1 using CHPF (note that CHSY3 = 'ChSy-2'). Izumikawa et al. (2008)

were apparently able to pull down CHSY1 and CHPF using CHPF2, and CHPF2 using CHSY3 (note that CHPF2 = 'ChSy-3').

9. The currently accepted idea seems to be that CHSY1/3 forms a heterodimer with CHPF(2) (e.g. Petit et al 2021 <https://doi.org/10.1093/glycob/cwaa086>)

10. Sammon et al (2023) already found that 'CS polymerase subunits could not be expressed alone; the only combination that produced sufficient soluble protein for enzymological analysis was CHSY3/CHPF.'

11. Sammon et al (2023) appear to have already developed an assay for monitoring the activity of purified CHSY3/CHPF on a similar substrate as the one used here.

Our answer: We thank the reviewer for the constructive and detailed feedback. In response, we have thoroughly revised and expanded the Introduction to incorporate the suggested information. Specifically, three additional paragraphs have been added, addressing the relevant background and context. These new sections are highlighted in yellow in the revised manuscript.

Results:

Fig. 1:

- Fig. 1a: Please state how this experiment was performed. Is this a Western blot of the culture supernatant or material bound to an affinity column? Is it possible that, when expressed individually, some proteins exhibit secretion defects and aggregate intracellularly?

Our answer: Western blot experiments were performed directly on the cell culture medium, and a corresponding statement has been added to the caption of Figure 1a. To further investigate why certain proteins were not detected in the medium during our co-expression assays, we also performed Western blot analysis on the corresponding cell pellets. These results are presented in an additional paragraph in the manuscript and shown in the newly added Supplementary Figure 1.

*"We further investigated intracellular protein expression to determine whether the absence of secreted protein was due to a lack of expression or inefficient secretion. In addition to the correctly secreted heterodimeric complexes (CHSY1–CHPF, CHSY1–CHPF2, CHSY3–CHPF, and CHSY3–CHPF2), we observed bands indicative of intracellular CHPF and CHPF2 expression (**Supplementary Fig. 1a, b**). A possible explanation for the intracellular retention of CHPF/CHPF2 when expressed individually is that they either fail to pass endoplasmic reticulum (ER) quality control or form heterodimeric complexes with endogenous, Golgi-localized CHSY1 or CHSY3."*

- Fig. 1b: The AlphaFold predicted heterodimers lack their N-terminal transmembrane anchors. Since some of these anchors are predicted to contain glycoprotein A-like dimerization motifs, how could one exclude that dimerization of the TM helices could give rise to another level of oligomerization? Could this be tested experimentally?

We have also performed AlphaFold predictions using the full-length amino acid sequences of the CS polymerase enzymes (see Response Letter Fig. 1). The N-terminal transmembrane anchoring helices exhibit very high predicted alignment error (PAE) values, indicating low confidence in the structural predictions for this region. To avoid overinterpretation and ensure clarity of visualization, we have opted to retain the original Figure 1b in the manuscript.

Fig. 1: AlphaFold prediction of full-length heterodimeric CS polymerase complexes

To further investigate whether a Glycophorin A-like motif could promote oligomerization, we analyzed the N-terminal region of the CHSY3–CHPF complex in more detail. AlphaFold predictions suggest the presence of a coiled-coil interaction in this region (see Response Letter Fig. 2).

Fig. 2: Superposition of the AlphaFold-predicted structure of CHSY3–CHPF with Glycophorin A

The N-terminal helices of CHSY3 and CHPF superimpose well with those of Glycophorin A, and both proteins contain the canonical GxxxG motif. However, because the AlphaFold predictions for the N-terminal regions are associated with low confidence, these findings would require further experimental validation - for example, by determining a high-resolution cryo-EM structure of the full-length CS polymerase complex embedded in a lipid nanodisc or detergent micelle. To further explore whether dimerization of the N-terminal membrane-anchoring helices could promote the formation of higher-order oligomeric complexes, we generated AlphaFold predictions of a CHSY3–CHSY3–CHPF trimer and a CHSY3–CHSY3–CHPF–CHPF tetramer (Response Letter Fig. 3). However, these predictions failed to yield meaningful structural models, suggesting that AlphaFold may not reliably capture potential higher-order oligomerization in this system.

Fig. 3: AlphaFold structure prediction of trimeric and tetrameric CS polymerase complexes

Fig. 2c and similar FACE results throughout:

- The gels require a molecular weight marker to indicate mass changes and to allow cross comparisons with gels shown in different figures. Further, the gels are shown at an inappropriate contrast that hides weaker bands. These are visible from the source data after adjusting the contrast. The presence of impurities in the synthesized substrate should be acknowledged and considered when interpreting the potential products of in vitro synthesis reactions.

Our answer: To improve comparability between gels and facilitate interpretation of the results, we repeated the FACE analysis using defined molecular weight markers. Specifically, we selected chemo-enzymatically synthesized penta-, hexa-, and heptasaccharide peptides, which were validated by mass spectrometry (Supplementary Fig. 5), as size standards. Figures 3, 5, and Supplementary Fig. 15 have been updated accordingly.

We would like to emphasize that no contrast adjustments were made to any of the FACE images. The high contrast observed is inherent to the use of the fluorophore (TAMRA). In

some gels, trace amounts of impurities (<3%) are visible; these stem from the TAMRA-peptide synthesis and the chemo-enzymatic linker addition. We believe these minor impurities do not affect the interpretation or validity of our results.

Fig. 3 and CHSY3-CHPF structure:

- For the resolution estimate of 3A it should be stated that this is only a nominal resolution estimate. The map quality of the N-terminal domain is closer to 3.5 to 4A over large regions.

Our answer: Ok, done as suggested.

- The authors do not discuss at all the architecture and potential function of the middle domain. Are there any structural homologs with known function?

Our answer: We thank the reviewer for these valuable suggestions. In response, we have added the following paragraph to the manuscript, along with three additional supplementary tables that summarize the results of the structural homology searches.

*"The middle domains of CHSY3 and CHPF are not believed to possess catalytic activity, but they play a key role in stabilizing the heterodimeric complex - e.g., through β -strand swapping that forms extensive and intricate hydrogen bonds between CHSY3 and CHPF. These middle domains are unique among glycosyltransferase structures, with the exception of the GalNAc transferases CSGALNACT1 and CSGALNACT2, which also participate in chondroitin sulfate biosynthesis⁴³. Structural similarity searches using the DALI server⁴⁴ (**Supplementary Table 5 and 6**) and Foldseek-Multimer⁴⁵ (**Supplementary Table 7**) primarily identified cysteine protease inhibitors and affimers as the closest structural homologs. However, the low sequence identity and lack of conserved catalytic residues argue against a shared biological function. Based on the current data, the middle domains likely serve primarily to stabilize the CS polymerase complex."*

- On several occasions, the authors refer to UDP as a donor substrate. UDP is not a substrate but a product of the enzymatic reaction (and perhaps an inhibitor). This should be distinguished.

Our answer: We modified the corresponding statements in the text.

Further, the authors do not discuss any potential donor or acceptor binding sites (for example for GalNAc, GlcA, or the oligosaccharide). A potential oligosaccharide acceptor was added to the cryo EM mix but it is not stated whether it was resolved in the EM map.

Our answer: The following statement has been added to the manuscript.

"No EM density was observed for the octasaccharide substrate that was added during sample preparation."

In the absence of experimental data, we chose not to speculate on the acceptor binding site and instead reserve a detailed description of the binding pockets for future studies.

- Lines 203: H394 is listed among residues that contact UDP, but it seems to be involved in cofactor (Mn²⁺) coordination.

Our answer: The statement has been modified accordingly and now reads: *"These include a DxD motif (D261 and D263), commonly found in GT-A fold enzymes^{39,40}, which, together with H394, coordinates the Mn²⁺ ion. Additional residues contributing to donor*

substrate binding in CHSY3 are M178, D231 and K238, located near the uracil moiety; D262, which interacts with the ribose group; and R187, Y233, K397, which stabilize the pyrophosphate group."

- In general, the novelty of the identified residues is overstated since most of them belong to conserved motifs that have been well characterized in other enzymes.

Our answer: Ok.

- Fig. 4a: Including a sequence alignment of, for example, CHSY3 and CHPF would be helpful to illustrate catalytically important residues that are absent in CHPF/CHPF2.

Our answer: Supplementary Figure 18b already includes a sequence alignment of all four CS polymerase proteins, highlighting the residues involved in UDP and Mn²⁺ binding.

- Fig. 4c and d: Molecular weight markers are required, and the images should be presented at an appropriate contrast level. Further, since both substrates were added to the reaction mix, it is important to characterize the products obtained, perhaps by MS/MS. Otherwise, it is difficult to determine what products have been formed, in particular considering the impurities present in the synthesized acceptor peptide.

Our answer: The FACE analysis gels shown in Fig. 5c, d (after figure renumbering) were repeated using penta-, hexa-, and heptasaccharide peptides as size markers, enabling unambiguous identification of the reaction products. Minor impurities are at the detection limit and do not affect the interpretation of the results.

- Fig. 4e: It should be acknowledged that the preference for Mn²⁺ has already been established. For this comparison, it would be informative to include a control in which no additional cation and no EDTA was added. This would indicate whether any cations copurify with the enzyme.

Our answer: We have added a statement acknowledging that the influence of metal ions has been investigated in previous studies. Additionally, we repeated the experiment to include a control reaction in which neither divalent cations nor EDTA was added.

"Analysis of the CHSY3-CHPF complex in the presence and absence of various metal ions revealed that both GlcA-T and GalNAc-T activities are metal ion dependent. The highest catalytic activity was observed with Mn²⁺, consistent with previous reports²⁸, while an intermediate activity was observed with Mg²⁺ and Ca²⁺ (Fig. 5e, f). GlcA-T and GalNAc-T activities were still observed in control reactions without added divalent cations. This finding suggests that the purified enzyme complex already has metal ions bound, likely originating from the buffer used during the initial purification steps."

- Another disease related mutation of CHSY3 (P626R) could be added to the analysis.

Our answer: We appreciate the reviewer's thoughtful suggestion. While we agree that the proposed experiment could provide some additional insights, it would require substantial time and resources. We believe the overall conclusions are sufficiently supported. We hope the reviewer understands our decision not to pursue this additional analysis at this stage.

Minor points:

- Introduction: For a general reader, please explain what is meant by transferase-II. What does 'II' refer to?

Our answer: We thank the reviewer for pointing this out. As the numbering does not add meaningful value in this context, we have removed it from the revised manuscript.

- Fig. 2b and corresponding text (lines 121-124): Annotating a single inflection point for the melting of the entire complex is misleading, in fact, if CSHY1-CHPF complexation turns out to be weaker, the negative inflection point seen around 50-55°C might be the actual quaternary structure melting point instead. The authors may simply state that the overall melting profile seem shifted towards a higher temperature in case of CSHY3-CHPF complex.

Our answer: We have now included the raw fluorescence traces of the $F_{350\text{nm}}/F_{330\text{nm}}$ ratio for both wild-type and mutant CS polymerase complexes as new Supplementary Figure 3a and 12b. These data support that the major unfolding transitions correspond to the peak maxima observed in the first derivative plots. While we do not speculate on the specific molecular events underlying the changes in fluorescence, we believe that reporting the melting temperatures is informative, as it allows readers to appreciate the range of temperature differences observed between the variants.

- Please avoid using 'electron' density when referring to EM map.

Our answer: We thank the reviewer for pointing this out. The corresponding text passages have been revised accordingly in the manuscript.

- -The description of UDP/Mn binding to the GT31 domain in lines 171-172 is repeated almost immediately in lines 199-200

Our answer: The statement was revised to avoid repetition.

- Line 201: The authors should specify that the DxD motif is a common motif of GT-A enzymes, not all GTs. Further, reference 36 seems inappropriate here. A better reference would be Wiggins & Munro 1998 (<https://doi.org/10.1073/pnas.95.14.7945>).

Our answer: We thank the reviewer for pointing this out. The corresponding text passages have been revised accordingly in the manuscript.

- Line 208 '4-β-N' should presumably be 'β-1,4-N'

Our answer: Ok, modified accordingly.

- According to the glycobiology nomenclature, glycans should be represented with their non-reducing end to the left and reducing end to the right. The authors may want to consider this when representing the glycopeptide sequences in the various figure panels.

Our answer: We thank the reviewer for pointing this out. All figures were revised accordingly.

- CHSY3-CHPF glycosylation sites: Some glycosylation sites reveal fairly strong densities beyond the modeled GlcNAc sugar. It would be informative to include an additional supplemental figure that shows these densities. Also, there may be an additional glycosylated residue in CHPF (N361).

Our answer: Although four N-glycosylation sites were identified in the CHSY3-CHPF complex (CHSY3: N279 and N710; CHPF: N138 and N361), clear EM density was observed for only three of them. The density at CHPF N361 was weak and insufficient for confident modeling. At the remaining three sites, a single GlcNAc residue could be modeled, but attempts to add further sugar moieties—such as a second GlcNAc, a fucose at CHSY3 N710, or a mannose at CHPF N138—were unsuccessful due to poor or

ambiguous density. Given the low confidence in these additional residues, we chose not to include them in the final structural model. As suggested, Supplementary Figure 10 has been added to illustrate the observed N-glycosylation sites.

- Line 428: It is suggested that all *four* CS synthases might have had activity early in evolution. Firstly, the divergences of CHSY1 from CHSY3 and CHPF from CHPF2 only appear to have occurred in the vertebrate lineage (Petit et al 2021 <https://doi.org/10.1093/glycob/cwaa086>). A single CHSY and CHPF descendant is found in e.g. the sponge Amphimedon. It would be relatively easy to test the hypothesis above by analyzing their sequences as well as those in other distantly related orthologues.

Our answer: We appreciate the reviewer's suggestion and have adjusted the statement accordingly to ensure a more precise and careful wording.

"Furthermore, it is conceivable that the ancestral proteins of CHPF (which gave rise to CHPF and CHPF2) and CHSY (which gave rise to CHSY1 and CHSY3)³³ initially possessed catalytic activity."

- Is there any information on the tissue specific expression levels of CHSY1/3/CHPF/2?
Our answer: The statement about tissue-specific expression of CS polymerase proteins has been revised and now reads:

"One possible reason for the existence of multiple CS polymerase complexes could be the tissue-specific expression patterns of the CS biosynthetic genes. For example, CHSY1 is more broadly expressed across tissues, whereas CHSY3 appears to be predominantly expressed in neuronal and glial cells."

- Supplementary Fig. 4: Can the molecular weights in panels b and c be indicated above each peak? It is important to show the individually observed masses in each spectrum so that the assignments can be verified in a case-by-case basis.

Our answer: Ok, the supplementary Figure 5 was updated accordingly.

- Supplementary Fig. 12: Please indicate what total cell number was used to normalize the data shown in panels b and c. Was this the area boxed in green and red on the right side of panel a?

Our answer: The total number of cells includes all measured events, not only those within the red and green boxes. Information about the total number of cells used for analysis (approximately 30,000) has been added to the Methods section. Precise numbers can be found in the Source Data 8.

- -Line 486: in which medium were the cells cultured? This is important for reproducibility as some media contain EDTA (which could interfere with nickel affinity chromatography).

Our answer: The missing information on cell medium was included in the revised version of the manuscript.

- Typos: Line 61 'trough', line 547 'serie', and line 580 'spectrometry'

Our answer: Ok, corrected.

- Line 25 and elsewhere: 'Distributive' and not 'disruptive' mechanism.

Our answer: Ok, corrected.

Reviewer #3 (Remarks to the Author):

The revised manuscript by Dutta et al. has improved significantly. The authors have addressed most of our concerns. However, one issue remains that should be addressed, either experimentally or as an additional discussion.

In the previous review, we suggested considering dimerization of the transmembrane helices via the glycophorin A motif as an additional or alternative mode of complex formation. The authors state in the rebuttal that AlphaFold predictions resulted in very low confidence in dimerization of this motif and thus dismiss the idea. It seems, however, that the opposite is the case.

First, the provided PAE plot for the CHPF/CHSY3 interaction shows a small area of high confidence (blue to white) for the transmembrane region.

Second, AlphaFold predicts the transmembrane helix dimerization with high confidence when using truncated CHSY3 and CHPF constructs. In fact, this is the only region that is predicted with high confidence in C-terminally truncated constructs.

Third, as found by a literature search, the transmembrane helix of CHPF2 (Uniprot: Q9P2E5) has previously been predicted by the CATM algorithm to form the strongest glycine zipper interaction of any protein in the entire human protein (Anderson et al 2017, JACS <https://doi.org/10.1021/jacs.7b07505>)

Considering these points, we believe that the possibility of homo- and/or heterodimerization mediated by the glycophorin A motifs should be experimentally tested (ideally) or at least discussed.

Our response: We thank reviewer #3 for drawing our attention to previous analyses suggesting a strong interaction between the transmembrane helices of CHPF2. While direct experimental testing of these interactions is technically very challenging and beyond the scope of this study, we agree that the manuscript would benefit from a careful discussion of the potential contribution of the N-terminal helices to oligomer formation.

To address this, we have added a new Supplementary Figure 19 showing the AlphaFold2 predictions for full-length heterodimeric CS polymerase enzymes. In addition, we include a superposition of the glycophorin A transmembrane coiled-coil with the CHSY3–CHPF complex. The corresponding Methods section has been updated accordingly.

The following paragraph was added to the manuscript:

" The contribution of the N-terminal anchoring helices to promoting oligomerization remains an important question. AlphaFold2 predictions of the full-length CS polymerase complexes suggest a potential coiled-coil formation between the N-terminal helices of CHSY3–CHSY1, CHSY3–CHPF, and CHSY3–CHPF2. Interestingly, the CHSY3–CHPF interface contains a glycophorin A-like motif (GxxxG), which is commonly observed in transmembrane helix interactions (Supplementary Fig. 19). "

Additional minor points:

Line 59: '4-β-N' should be 'β-1,4-N' Corrected.

Fig. 3c: which combination of enzymes produced the chondroitin product used to test for chondroitinase sensitivity? The figure legend now reads: "Analysis of CHSY3-CHPF reaction product upon chondroitinase ABC treatment."

Page 9: the ordering and correspondence of supplementary figures to their references seems to be messed up here. We thoroughly corrected the numbering of supplementary figures.

Line 222: typo 'PSIA' (should be 'PISA') Corrected.

Line: 265: Can M178 really be said to interact with the UDP uracil moiety? It is 6.2 Å from the uracil and 5.5 Å from the ribose. M178 was removed from the list of residues proposed to be involved in UDP coordination.